# Genomic and transcriptomic variation defines the chromosome-scale assembly of *Haemonchus contortus*, a model gastrointestinal worm

Stephen R. Doyle ⓘ et al.#

*Haemonchus contortus* is a globally distributed and economically important gastrointestinal pathogen of small ruminants and has become a key nematode model for studying anthelmintic resistance and other parasite-specific traits among a wider group of parasites including major human pathogens. Here, we report using PacBio long-read and OpGen and 10X Genomics long-molecule methods to generate a highly contiguous 283.4 Mbp chromosome-scale genome assembly including a resolved sex chromosome for the MHco3(ISE).N1 isolate. We show a remarkable pattern of conservation of chromosome content with *Caenorhabditis elegans*, but almost no conservation of gene order. Short and long-read transcriptome sequencing allowed us to define coordinated transcriptional regulation throughout the parasite's life cycle and refine our understanding of *cis*- and *trans*-splicing. Finally, we provide a comprehensive picture of chromosome-wide genetic diversity both within a single isolate and globally. These data provide a high-quality comparison for understanding the evolution and genomics of *Caenorhabditis* and other nematodes and extend the experimental tractability of this model parasitic nematode in understanding helminth biology, drug discovery and vaccine development, as well as important adaptive traits such as drug resistance.

---

# A list of authors and their affiliations appears at the end of the paper.

The recent introduction of long-read DNA sequencing from Pacific Biosciences (PacBio) and Oxford Nanopore has completely changed the quality and contiguity of assemblies of both large and small genomes. The ability to generate highly contiguous and closed prokaryote genomes is becoming routine, even from metagenomic samples containing multiple strains or species[1,2], and high-quality chromosome-scale genomes for many eukaryotic species are rapidly becoming available[3–5]. These technologies generally require microgram quantities of high molecular weight DNA, and the success of a genome assembly is highly dependent on the degree of polymorphism present in the DNA that is sequenced. Ideally, the DNA sequenced will be obtained from a single individual—so that the only polymorphism is heterozygosity between homologous chromosomes in the case of a diploid or polyploid individual—or from a clonal population, where multiple but genetically identical individuals can be pooled. However, for many small organisms, this is not possible; the only option is to sequence DNA from a pool of genetically distinct organisms and derive a consensus assembly. As modern assembly algorithms are designed and trained on haploid or diploid genomes from single individuals[6–8], the excessive diversity presents a significant challenge to the assembly process, and consequently, typically results in fragmented assemblies that contain misassembled and excessive haplotypic sequences[9].

Parasitic helminths are a diverse group of organisms for which significant efforts have been made to develop genomic resources[10]. For some species[11–16], high-quality near-complete assemblies have been achieved using multiple complementary sequencing and mapping technologies, and usually with extensive manual improvement. Similar quality assemblies for a few other species are available from WormBase ParaSite[17], but await formal description in the peer-reviewed literature. However, for many others, access to sufficient high-quality DNA, small organismal size, and high genetic diversity are significant challenges that need to be overcome to achieve highly contiguous assemblies. The nematode *Haemonchus contortus* is one such species. *H. contortus* is a major pathogen of sheep and goats worldwide and is recognised as a model parasite due to its experimental tractability for drug discovery[18,19], vaccine development[20,21] and anthelmintic resistance research[22]. Draft genome assemblies were published in 2013 for two anthelmintic-sensitive isolates, MHco3 (ISE).N1[23] and McMaster[24]; both assemblies were produced using short-read, high-throughput sequencing technologies (including both short- and long-insert libraries), resulting in assembly sizes of 370 Mbp and 320 Mbp, respectively. However, direct comparison of these assemblies shows discordance, revealing clear differences in gene family composition[25] and variation in assembly quality and gene content[26]. Although some of these differences reflect biological variation in this highly polymorphic species[27–29], many differences are technical artefacts, as revealed by comparing fragmented genome assemblies from sequencing DNA derived from pools of a genetically diverse organism. These fragmented genome assemblies are less than ideal for many downstream uses of reference genomes, for example, in genetic experiments in which patterns of genome-wide variation are defined and interpreted[30].

Here we present the chromosome-scale assembly of the MHco3(ISE).N1 isolate of *H. contortus*, representing the first high-quality assembly for any strongylid parasite, an important clade of parasitic nematodes that includes parasitic species of veterinary and medical importance. Using a hybrid assembly approach incorporating short- and long-read sequencing followed by manual finishing, we demonstrate how a highly contiguous assembly overcomes some of the critical limitations imposed by draft assemblies in interpreting large-scale genetic and functional genomic datasets. We provide insight into patterns of transcriptional change and putative co-regulation using a significantly improved genome annotation, derived *de novo* from both short- and long-read cDNA sequencing, and describe within- and between-population genetic diversity that shapes the genome. This chromosome-scale assembly now offers insight into genome evolution among a broad group of important parasite species and a robust scaffold for genome-wide analyses of important parasite traits such as anthelmintic resistance.

## Results

**Chromosome structure of *Haemonchus contortus*.** We have built upon our previous assembly (version 1 (V1)[23]) using a hybrid approach, iteratively incorporating Illumina short-insert and 3 kbp libraries, PacBio long-read, OpGen optical mapping, and 10X Genomics linked-read data (Supplementary Fig. 1; Supplementary Data 1) to generate a largely complete, chromosomal-scale genome assembly. Consistent with the karyotype for *H. contortus*[31,32], the assembly consists of five autosomal scaffolds and one sex-linked scaffold (Supplementary Fig. 2), each containing terminal telomeric sequences (sequence motif: TTAGGC), and a single mitochondrial contig. We assigned chromosome names based on broad-scale synteny with *C. elegans* chromosomes (Fig. 1a); over 80% of 7,361 one-to-one orthologous genes are shared on syntenic chromosomes between the two species demonstrating the high conservation of genes per chromosome (Fig. 1b; top), however, vast rearrangements are evident and very little conservation of gene order remains (Fig. 1b; bottom). We determined the extent of microsynteny by comparing conserved, reorientated, and rearranged gene pairs between the two species (Supplementary Fig. 3a); the distance between ortholog pairs in *H. contortus* and *C. elegans* is correlated up to ~100 kbp (rho = 0.469, $P = 2.2E-16$) in which pair order is conserved (Supplementary Fig. 3b), likely representing selective constraint to maintain evolutionarily conserved operons, but is largely lost above 100 kbp (rho = 0.017, $P = 0.660$) where a greater frequency of rearrangement between pairs is evident (Supplementary Fig. 3b, c). Beyond pairs of genes, synteny breaks down rapidly; although almost 50% of shared ortholog pairs are adjacent to each other, only a single group of 10 orthologs are colinear between the genomes of the two species (Supplementary Fig. 3d, e; Supplementary Data 2).

The V4 assembly is 283.4 Mbp in length (scaffold N50 = 47.3 Mbp; contig N50 = 3.8 Mbp), representing only 75.6% of the 369.8 Mbp V1 draft assembly length (Table 1); the reduction in assembly size is due to the identification and separation of redundant haplotypic sequences present in the polymorphic draft V1 and preliminary PacBio assemblies generated. The V4 assembly is highly resolved with only 185 gaps, a substantial reduction from the 41,663 gaps present in the V1 assembly (Table 1). We identified 242 of 248 (97.98%) universally conserved orthologs measured by the Core Eukaryotic Genes Mapping Approach (CEGMA; Supplementary Table 1), and 859 of 982 (87.4%) metazoan Benchmarking Universal Single-Copy Orthologs (BUSCOs). The remaining missing orthologs show phylogenetic structure among clade V nematodes (Supplementary Fig. 4), suggesting that many are truly missing from the genomes of these species rather than due to assembly artefacts. The V4 assembly contained 141 more full-length single-copy orthologs and 159 fewer duplicated BUSCOs than identified in the V1 genome (Supplementary Table 1); considering that we identified a similar average number of CEGMA orthologs per core gene in V4 and in the *C. elegans* genome (1.1 in V4, compared with 1.09 in *C. elegans*), erroneously-duplicated sequences have been reduced in V4 compared with the draft *H. contortus* assemblies (V1: 85.89%

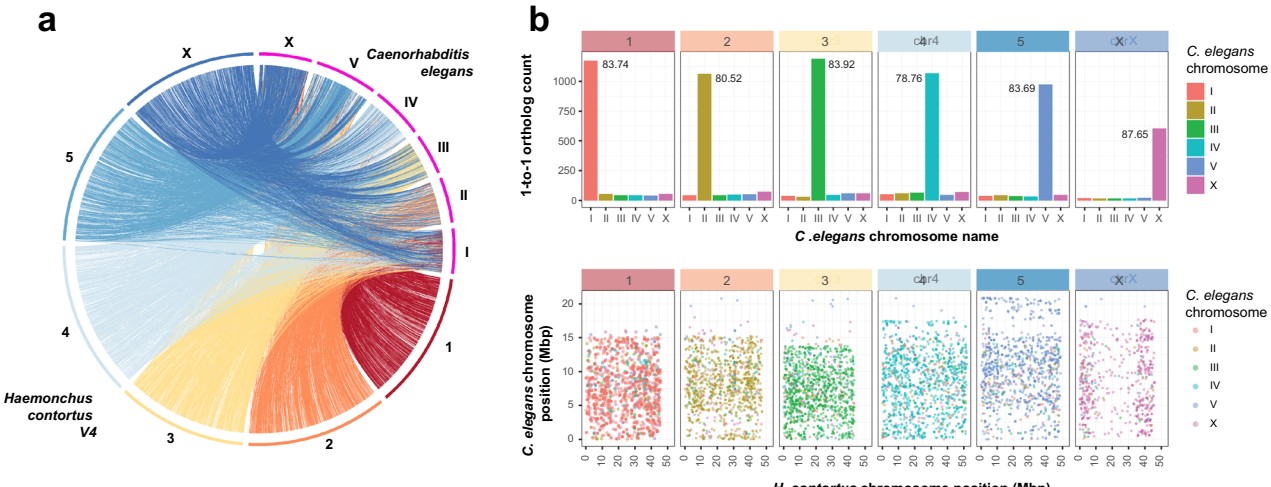

**Fig. 1 Chromosomal synteny between *Haemonchus contortus* and *Caenorhabditis elegans*. a** Genome-wide comparison of chromosomal organisation between *H. contortus* V4 and *C. elegans*. **b** Comparison of shared orthologs, based on assignment to chromosomes (top plot) and their relative position on the chromosome (bottom plot). In the top plot, chromosome assignment of 7,361 one-to-one orthologs between *C. elegans* and *H. contortus* V4 annotations demonstrates that the majority of genes on any given *H. contortus* chromosome are found on the same chromosome in *C. elegans*. Values within each panel represent the percentage of shared orthologs found on the same chromosome of both *H. contortus* and *C. elegans*. In the bottom plot, the relative genomic position of the one-to-one orthologs between *C. elegans* (*y*-axis) and *H. contortus* V4 chromosomes (*x*-axis) is shown. Hypothetically, if *C. elegans* and *H. contortus* chromosomes were completely colinear, the genomic positions of orthologs would show a positive linear relationship between the two species; however, this is not the case, whereby an almost complete reshuffling of orthologs was observed.

**Table 1 Genome assembly summary statistics.**

|  | *H. contortus* V4 Chromosomes | *H. contortus* V4 Haplotypes | *H. contortus* V1[a] | *H. contortus* McMaster[b] | *H. contortus* New Zealand[c] | *C. elegans* WB[d] |
|---|---|---|---|---|---|---|
| Genome size (bp) | 283,439,308 | 248,771,548 | 369,846,877 | 319,640,208 | 465,720,674 | 100,286,401 |
| Scaffolds | 7 | 3,907 | 23,860 | 14,419 | 7 | 7 |
| Scaffold N50 (bp) | 47,382,676 | 167,270 | 83,287 | 56,328 | 83,970,805 | 17,493,829 |
| Scaffolds ≥N50 | 3 | 174 | 1,151 | 1,684 | 3 | 3 |
| Contigs | 192 | 4,226 | 65,523 | 55,322 | 172,899 | 7 |
| Contig N50 (bp) | 3,801,457 | 117,552 | 20,808 | 12,900 | 3,087 | 17,493,829 |
| Contigs ≥N50 | 20 | 439 | 4,943 | 6,646 | 43,632 | 3 |
| Scaffold N content (bp) | 5,699,711 | 14,755,378 | 23,804,39 | 20,495,720 | 17,288,295 | 0 |
| Scaffold gaps | 185 | 323 | 41,663 | 40,903 | 172,892 | 0 |

[a]Haem V1[23]: haemonchus_contortus.PRJEB506.WBPS10.genomic.fa
[b]McMaster[24]: haemonchus_contortus.PRJNA205202.WBPS11.genomic.fa
[c]New Zealand[33]: ENA accession GCA_007637855.2: GCA_007637855.2_ASM763785v2_genomic.fna
[d]*C. elegans* WBcel235: caenorhabditis_elegans.PRJNA13758.WBPS9.genomic_softmasked.fa.

complete and 1.59 average orthologs per CEGMA gene; McMaster: 70.56% complete and 1.43 average orthologs CEGMA gene; Supplementary Table 1).

An assembly of a New Zealand (NZ) *H. contortus* strain was recently made available[33], which used our V4 chromosomal assembly to scaffold a highly fragmented draft assembly (contigs = 172,899) into seven chromosome-scale scaffolds. The expansion of the NZ assembly size (465 Mbp vs 283 Mbp in the V4 assembly; Table 1) in the context of elevated duplicated BUSCOs (7.3% vs 1.6% in the V4 assembly) and the increased proportion of orthologs detected, on average, per partial and complete CEGMA gene (1.42 and 1.71, respectively, vs 1.1 and 1.22 in the V4 assembly; Supplementary Table 1) suggest that contaminating haplotypic sequences have erroneously been incorporated into the NZ assembly. Similarly, we suspect that the lower CEGMA statistics together with the reported low average exon number (4 exons per gene vs 9 exons in the V4 assembly) and transcript lengths (666 bp vs 1,237 bp in the V4 assembly; see below) suggest that frameshift errors due to insertion and/or deletions from uncorrected PacBio reads and/or assembly

gaps (*n* = 172,892) are affecting coding sequences broadly throughout the NZ assembly. While multiple chromosome-scale assemblies from the same species provide a unique opportunity for comparative analysis, the striking differences between the NZ and the V4 assemblies are unlikely to represent genuine biological variation between strains.

**Resolving haplotypic diversity and repeat distribution within the chromosomes**. We identified a dominant group of haplotypes in our PacBio assemblies which, together with Illumina read coverage derived from a single worm, was used to partially phase the chromosome assembly sequence by manually curation. Mapping of the additional haplotypes to the reference-assigned haplotype (Fig. 2a) highlighted both the diversity between the haplotype and chromosomally-placed sequences (Fig. 2b, c second circle) and coverage of the haplotypes, spanning approximately 66.8 ± 24.1% on average of the chromosome sequences (Fig. 2c third circle; range 85.1 ± 12.3% [chromosome I] to 51.8 ± 16.6%

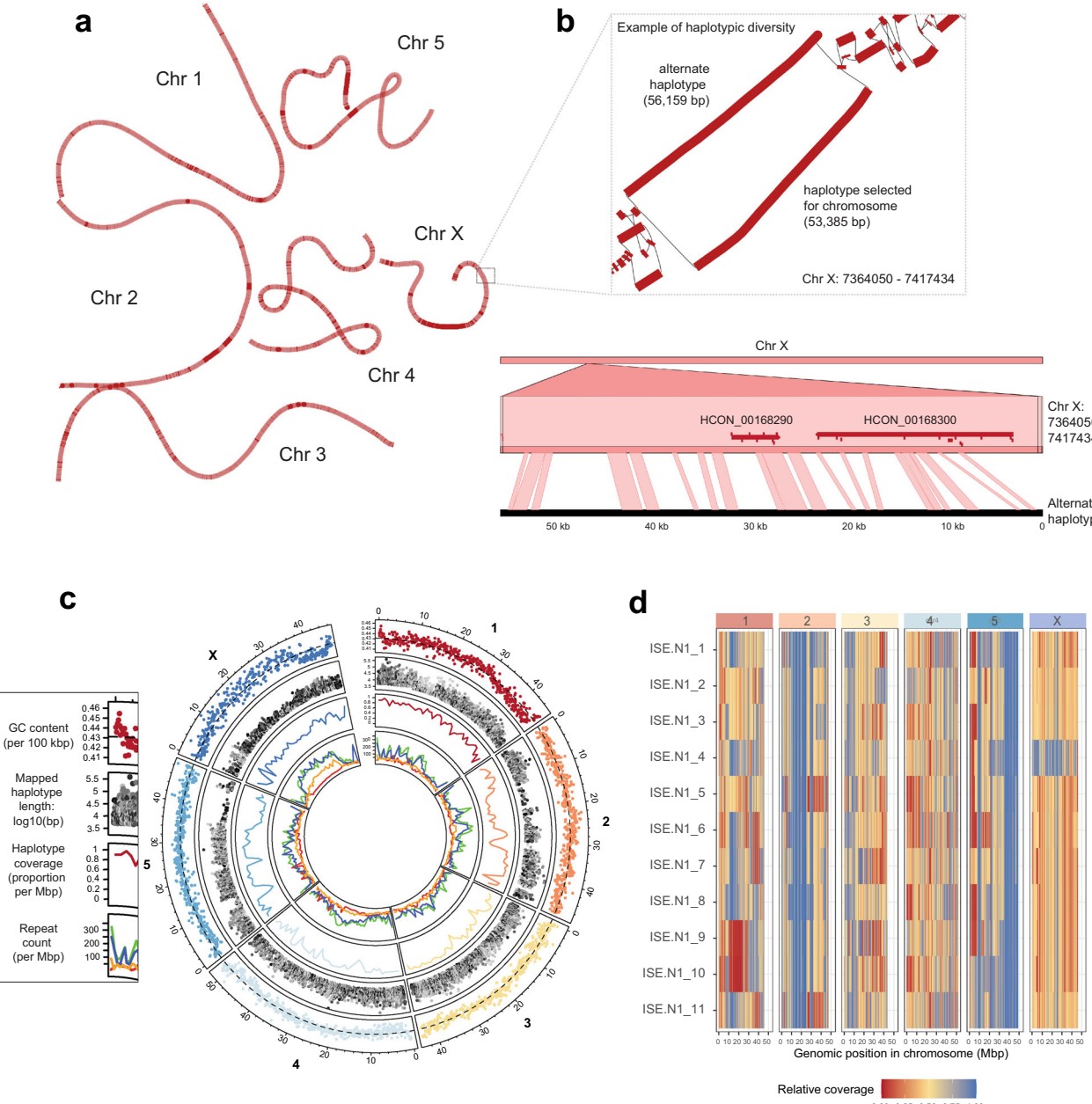

**Fig. 2 Haplotype and repeat diversity within and between chromosomes. a** Genome graph derived from the whole-genome alignment of chromosome and alternate haplotype assemblies. The graph is coloured to reflect alternative haplotype density, distinguishing regions with longer reference only-haplotypes (dark red) from those with shorter, more diverse haplotypes (light red). **b** A close-up view of the genome graph structure (box) highlights an example of alternate haplotypes from 7.36 to 7.42 Mbp on Chr X; alternative haplotype blocks are represented by thick red lines and paths between haplotypes as thin black lines. DNA alignment of the 56 kbp alternate haplotype to the genome demonstrates the extent of the genetic diversity between these sequences; in this region, two genes are present (red horizontal bars), of which their exons (red ticks) are predominantly found in regions of conservation (ribbons connecting the top and bottom sequences). **c** Circos plot of genome-wide haplotype and repeat diversity. Outer: mean GC nucleotide frequency in 100 kbp windows. The dashed line represents genome-wide mean GC content (0.429); Second: Distribution (x-axis) and length (y-axis) of mapped haplotype sequences to the chromosome, with sequence similarity between the mapped haplotype and chromosomes indicated by the grey-scale gradient (high similarity = dark, low = light); third: proportion of chromosome covered by a least one haplotype, measured in 1 Mbp windows; inner: density of annotated repeats, including LINEs (green), SINEs (red), DNA (blue), and LTRs (orange). **d** Haplotype switching between the chromosome scaffolds and additional haplotypes present in the genome assembly. Whole-genome sequencing was performed on individual worms (n = 11; MHco3(ISE). N1 strain, used in the genome assembly), after which reads were mapped competitively against the chromosomes and haplotype sequences. Normalised read depth coverage is presented, which was calculated per 100 kbp relative to the maximum genome-wide coverage set at the 95% quantile to prevent high-coverage outlier regions from skewing coverage estimates. Blue = ~100% of maximum coverage; Yellow = ~50% of maximum coverage; Red = ~0% of maximum coverage.

[X chromosome]). Whole-genome short-read sequencing of 11 single worms (10 male and 1 female, identified by full- or half-coverage across the X chromosome for XX females and XO males, respectively) followed by competitive read mapping revealed clear evidence of haplotype switching between the chromosomal reference and alternate haplotype assembly based on differences in the sequencing coverage throughout the genome and between samples (Fig. 2d). Haplotypic blocks are evident, including large regions on chromosome 2 and 5 that are devoid of haplotype diversity (full coverage of the reference; blue), as well as regions in which individuals lack any read coverage due to the absence of the reference haplotype in their genome (e.g. red region on chromosome 1, samples 09 & 10). These data suggest that, despite being semi-inbred through a limited number of single-pair matings[34], complex haplotypes are segregating in this population, and that the frequency and diversity of these haplotypes are non-randomly distributed throughout the genome.

The use of multiple long-molecule technologies allowed a greater representation of repetitive sequences in the V4 assembly, accounting for approximately 36.43% (compared with 30.34% in V1) of the genome (Supplementary Table 2). Particular repeat classes, including LINEs, LTRs and DNA elements were enriched towards the middle of the chromosomes and the chromosome ends (Fig. 2c inner circle; Supplementary Fig. 5). This distribution pattern is negatively correlated with observed recombination rate domains in both *H. contortus*[35] and *C. elegans*[36], suggesting that transposons accumulate in regions of low recombination rate throughout the genome. On the X chromosome, this enrichment is striking in the sub-telomeric regions and, in particular, from approximately 8 to 35 Mbp along the chromosome (Fig. 2c inner circle; Supplementary Fig. 5). Finally, the pattern of repetitive elements throughout the chromosomes and not the coding sequences (shown below) correlates with and perhaps is driving the nucleotide composition profile throughout the genome (Fig. 2c outer circle).

**Generation of a high-quality transcriptome annotation incorporating short and long reads**. Considering the broad-scale changes from V1 to V4 assemblies, we undertook a *de novo* gene finding strategy to annotate the V4 genome. To do so, we used our Illumina short-read RNA-seq collection sampled from seven stages of the parasite life cycle (described in the analysis of the V1 genome[23]) and supplemented this with PacBio Iso-Seq long-read sequencing (Supplementary Table 3) of full-length cDNA from adult male, female, and juvenile L3 parasites. The overall annotation strategy is presented in Supplementary Fig. 6 and consisted of four key stages: (i) RNA-seq evidence was used to train ab initio gene finding using Braker[37]; (ii) full-length high-quality isoforms generated using Iso-Seq were used to identify candidate gene models using the annotation tool Program to Assemble Spliced Alignments (PASA)[38]; (iii) evidence from steps (i) and (ii), together with 110 manually curated genes and orthologs from the *C. elegans* and *H. contortus* V1 genomes, were incorporated using EvidenceModeller; and finally (iv) UTRs and multi-evidence gene models were updated using the full-length isoforms once again using PASA. Together, this strategy achieved a high level of sensitivity and precision, and represented substantial improvement over the V1 annotation (Supplementary Table 4).

The V4 annotation comprises 19,489 nuclear genes encoding 20,987 transcripts with approximately 56.4% (23,687 of 41,974) of UTRs annotated. The increase in genome contiguity, together with the incorporation of full-length cDNA sequencing (Fig. 3a), has resulted in longer gene (40.5% increase), mRNA (41.9% increase), and exon (30.6% increase) models on average than the previous V1 genome annotation (Fig. 3b; Supplementary Table 5);

our approach allowed better representation of the longest genes (447,146 bp compared with 91,953 bp in V1), and intriguingly, identified 43 of the longest 50 (86%) and 73 of the longest 100 gene models on the X chromosome alone. These longest genes were predominantly located toward the centre of the X chromosome, suggesting that gene length expansion is driven by the accumulation of intronic repetitive elements enriched in this region as described above (Fig. 2c inner circle; Supplementary Fig. 5). Concordantly, both the gene number and gene coverage of 100 kbp windows were significantly lower on the X chromosome relative to the autosomes (pairwise t-test between X and autosomes; $P = 2.2E-16$). The reduction in haplotypic sequences together with greatly improved gene models resolved 61% more one-to-one orthologs with *C. elegans* ($n = 7,361$) than between *C. elegans* and V1 ($n = 4,529$) (Supplementary Table 6). Surprisingly, a greater number of one-to-one orthologs are identified between V4 and *Haemonchus placei* ($n = 9,970$; https://parasite.wormbase.org/Haemonchus_placei_prjeb509/Info/Index/) than V4 and V1 ($n = 9,595$), likely reflecting the higher duplication rate and haplotypic nature of the V1 relative to the *H. placei* assembly, although this also highlights the close relationship between *H. contortus* and *H. placei*[39], a gastrointestinal pathogen commonly associated with cattle. The increased orthology between close (*H. placei*) and more distantly (*C. elegans*) related species provides support for a more complete and biologically representative gene set in the V4 annotation than the previously described V1 and McMaster annotations.

**Transcriptional dynamics throughout development and between sexes**. Both 2013 publications describing draft assemblies of *H. contortus* provide extensive descriptions of the key developmental transitions between sequential pairs of life stages throughout the life cycle[23,24]. We revisited this using the current annotation and, for visual comparison, present the top 1,000 most variable transcripts across all life stage samples in the MHco3 (ISE).N1 datasets (Fig. 4a; Supplementary Data 3). The transcriptional transition between the juvenile free-living life stages and the more mature parasitic stages is striking, reflecting not only the development toward reproductive maturity but also drastic changes in the parasite's environment. To explore these developmental transitions more explicitly, we determined the co-expression profiles for all genes throughout the lifecycle, revealing 19 distinct patterns of co-expression containing 8,412 genes (41% of all genes), with an average cluster size of 442.7 genes (Supplementary Fig. 7; Supplementary Data 4). Consistent with the pattern of highly variable genes, two dominant clusters of genes were identified: (i) genes with higher expression in free-living stages that subsequently decreased in expression in parasitic stages (cluster 1; $n = 1,550$ genes; Fig. 4b) and (ii) genes with lower expression during free-living stages that subsequently increased in expression in parasitic stages (cluster 8; $n = 1,542$ genes). Functional characterisation of the cluster 1 genes revealed 45 enriched gene ontology (GO) terms (https://biit.cs.ut.ee/gplink/l/8jECkGwQQS) that predominantly described ion transport and channel activity, as well as transmembrane signalling receptor activity. For cluster 8, 22 GO terms were enriched (https://biit.cs.ut.ee/gplink/l/oJoq6Zh0Re) and included terms describing peptidase activity, as well as protein and organonitrogen metabolism. To provide further resolution, we tested *C. elegans* orthologs of this gene set, which revealed greater resolution on metabolic processes, including terms associated with glycolysis and gluconeogenesis, pyruvate and nitrogen metabolism (https://biit.cs.ut.ee/gplink/l/b3Bx8rhJR5). Finally, we explored whether co-expressed genes contained shared sequence motifs in their 5' UTR that may be indicative of true

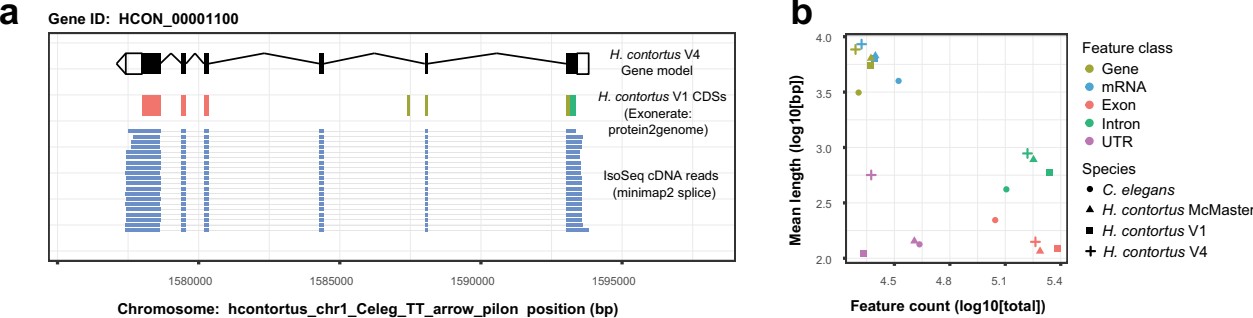

**Fig. 3 De novo transcriptome annotation incorporating full-length cDNA sequencing results in longer and more complete annotations. a** Example of an improved and now full-length gene model. *HCON_00001100*, a nuclear hormone receptor family member orthologous to *C. elegans nhr-85*, was incomplete and fragmented in the V1 assembly (multiple colours representing CDSs from three distinct V1 annotations, *HCOI00573500*, *HCOI00573600*, and *HCOI00573700*). The full-length gene model, consisting of six exons (black boxes) flanked by UTR sequences (white boxes) is supported by full-length cDNA Iso-Seq sequences (blue bars). **b** Comparison of mean length and abundance of annotation features between *H. contortus* V1 (square), *H. contortus* V4 (cross), *H. contortus* McMaster (triangle) and *C. elegans* (circle) transcriptome annotations.

transcriptional co-regulation; while broadly poorly represented, we did identify 70 enriched motif sequences of which 17 were associated with gene sets with enriched GO terms (Supplementary Data 5), for example, cysteine-type peptidase activity in cluster 8 (GATAAGR; motif E-value = 2.90E−02; GO $P$ = 4.181E −13), DNA replication in cluster 15 (AAAAATVA; motif E-value = 3.50E−02; GO $P$ = 6.269E−4), and molting cycle, collagen and cuticulin-based cuticle in cluster 16 (GTATAAGM; motif E-value = 2.30E−02; GO $P$ = 1.926E-6), suggestive of either direct or indirect co-regulation of expression.

The greatest differences in gene expression between life stages are between L4 (mixed male and females) and either adult males or adult females, or between adult males and adult females, with a strong sex-bias towards increased gene expression in adult males (Supplementary Data 3). Consistent with *C. elegans*, sex-biased expression is not randomly distributed throughout the genome[40,41]; only 5% (146/2,923) of *H. contortus* male-biased genes are found on the X chromosome (relative to 14.3% of all genes found on the X chromosome; $\chi^2$ = 175.83, df = 1, $P$ = 3.84E−40), whereas more female-biased genes are X-linked than expected (24.6% [110/476]; $\chi^2$ = 33.59, df = 1, $P$ = 6.78E−9). Comparison of sex-biased genes on the X chromosome revealed that, while almost completely absent in males, X-linked female-biased expression identified *C. elegans* orthologs previously associated with development, including developmentally upregulated genes (e.g., *smad4*, transcription factor AP2, *vab-3*), genes involved in egg development (e.g., *egg-5*, *vit-1*, *cpg-2*, *egg-1*), as well as genes involved in developmental timing (e.g., *lin-14*, *kin-20*[42]) that are expressed downstream of the sex-determination regulator *tra-1*. Sex determination in *C. elegans* is mediated by the ratio of X chromosomes to autosomes, whereby organisms with a single X chromosome will develop as a male (X-to-autosome ratio = 0.5:1, i.e., XO) and those with two X chromosomes develop as a hermaphrodite (XX; X-to-autosome ratio = 1:1); similarly, *H. contortus* males are XO and females are XX[31,32]. The detection of the X-to-autosome copy number initiates a regulatory pathway that first activates a dosage compensation response that down-regulates gene expression in hermaphrodites by one-half, followed by sex determination mediated by the conserved transcription factor *tra-1*; in XX individuals *tra-1* is activated, whereas *tra-1* is repressed in XO individuals. Although the similarities in the XX/XO chromosome structures between *C. elegans* and *H. contortus* suggest that the mechanisms controlling sex determination may be similar, this has not been previously shown. Comparison of *H. contortus* RNA-seq data between males and females showed no significant difference in the distribution

of expression between autosomes and X-linked genes; only 9.24% of genes (110 + 146 of 2,770 X-linked genes) are differentially expressed on the X chromosome between sexes suggests that X chromosome dosage compensation is active in XX females. To explore this further, we mapped *H. contortus* orthologs of *C. elegans* genes involved in the X chromosome sensing and dosage compensation response and subsequent sex determination pathway (Fig. 4c; Supplementary Data 5; Adapted from Worm-Book[43]). In *C. elegans*, chromosome dosage is determined by measuring the expression of specific genes on the X and autosomes; in *H. contortus*, only two of the *C. elegans* X-linked orthologs - *fox-1* and *sex-1* - are present, however, these are located on the autosomes, whereas the autosomal dosage associated genes *sea-1*, −2, and −3, are missing. Five of the six *C. elegans* orthologs associated with the dosage compensation complex (DCC) were found (*dpy-26* is missing), however, the genes *xol-1* and *sdc-1*, −2, and −3 that act to initiate the recruitment of the DCC are absent. Thus, while the DCC may still mediate dosage compensation in *H. contortus*, the mechanism used to determine the X-to-autosome ratio is likely different from *C. elegans*. Identification of many orthologs downstream of *her-1*, which receives X chromosome dosage signals to initiate sex determination, suggests greater conservation of this mechanism between species. Notable is the absence of *H. contortus* orthologs of *tra-2* and *fem-3*, as they are key members of the pathway that receive regulatory signals promoting male or female developmental pathways, respectively, in *C. elegans*.

**Transcriptome complexity is generated by extensive *cis*- and *trans*-splicing.** Spliced leader (SL) *trans*-splicing has been extensively described in *C. elegans*[44] and has been shown previously to operate in *H. contortus*[45–47] (Fig. 5a). To explore this further in the V4 annotation (Supplementary Fig. 8a, b), we identified 7,293 (37.4%) and 1,139 (5.8%) genes with evidence of SL1 and SL2 *trans*-splicing, respectively (Supplementary Data 7). However, additional evidence of *trans*-splicing was observed; 7,895 and 1,222 annotated transcripts contained SL1/ SL2 sequences, and a further 6,883 internal SL cut sites indicated additional non-annotated isoforms were present in the transcriptomes. While we likely underestimate the frequency of *trans*-splicing due to the modest RNA sequencing depth available, this approach reveals a higher frequency of *trans*-splicing than previously described and additional evidence for the prediction and annotation of novel gene isoforms. Further, we identify degeneracy in the nucleotide sequences of both SL1 and SL2 sequences (Supplementary Fig. 8c), and conservation in the genome

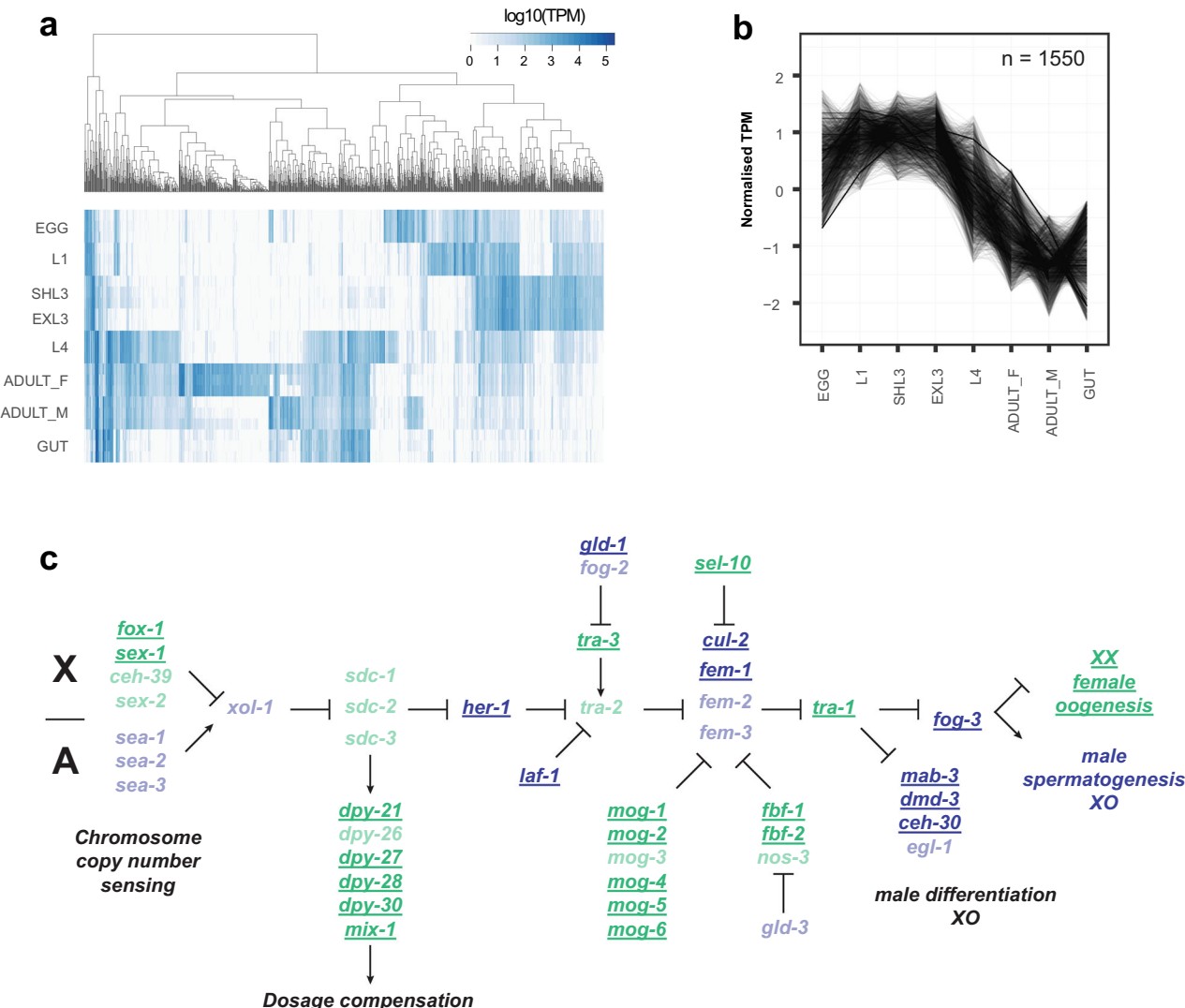

**Fig. 4 Transcriptional dynamics throughout development and between sexes. a** Heatmap of the most variable transcripts across the life stages. Transcript abundance measured as transcripts per million (TPM) was determined using kallisto, from which the TPM variance was calculated across all life stages. The top 1,000 most variable transcripts with a minimum of 1 TPM are presented. Genes differentially expressed in each pairwise comparison throughout the life cycle are presented in Supplementary Data 3. **b** Transcript co-expression profiles across life stages of *H. contortus*. Cluster 1, which contains the most co-expressed genes, is presented and broadly reflects the transition between juvenile free-living life stages and mature parasitic stages. The remaining profiles are presented in Supplementary Fig. 8, with genes by cluster described in Supplementary Data 4. **c** Schematic of the chromosome dosage sensing and sexual development pathway of *C. elegans*. Adapted from WormBook[43], we highlight the presence of *H. contortus* orthologs (bold, underlined) of *C. elegans* genes previously demonstrated to promote male (blue) or female (green) development. Missing orthologs from the *H. contortus* genome are translucent. A full list of *C. elegans* genes and *H. contortus* orthologs from the pathway are presented in Supplementary Data 6.

sequence immediately upstream of the splice site, particularly for SL2 (Supplementary Fig. 8d). In *C. elegans*, SL1 sequences are typically associated with non-operon genes or the first coding sequence of an operon, whereas SL2 sequences are used for second and subsequent coding sequences in a polycistronic operon. The intercistronic region between SL2-spliced genes and their upstream genes in *C. elegans* are typically short, usually approximately 50 to 200 bp on average, with the majority less than 2 kbp[44]. In *H. contortus*, the intergenic distance upstream of genes with SL1 sequences is largely consistent with that of genes without SL1 sequences (Fig. 5b left plot; median = 8,557 bp [SL1-spliced], 8,603.5 bp [no SL]); however, a small proportion of SL1-spliced genes are found less than 100 bp downstream of the nearest gene. Conversely, while a larger proportion of genes with SL2 sequences are immediately downstream of the nearest gene as expected (median distance = 566.5 bp), a substantial proportion

(432 [35.4%]) are found with a large intergenic distance (>2 kbp) to the nearest upstream gene (Fig. 5b right plot). These observations suggest that while broadly consistent with *C. elegans*, (i) *H. contortus* SL1-sequences can be found in downstream genes within a polycistron, and may reflect evidence of hybrid operons that contain a second, internal promoter to control the expression of a subset of genes within the operon[48], (ii) SL2-spliced genes are not necessarily in a polycistron but may function independently, and/or (iii) recognition of SL2 splicing within an operon remains effective over large physical distances, for example, the 2,064 bp region between *deg-3* (HCON_00039370) and *des-2* (HCON_00039380) operon of *H. contortus*[49]. Although these variations have been described in *C. elegans*[50], they seem to be more the norm rather than the exception in *H. contortus*.

We extended the splice variation analysis to characterise *cis*-splicing of transcripts and differential isoform usage between life

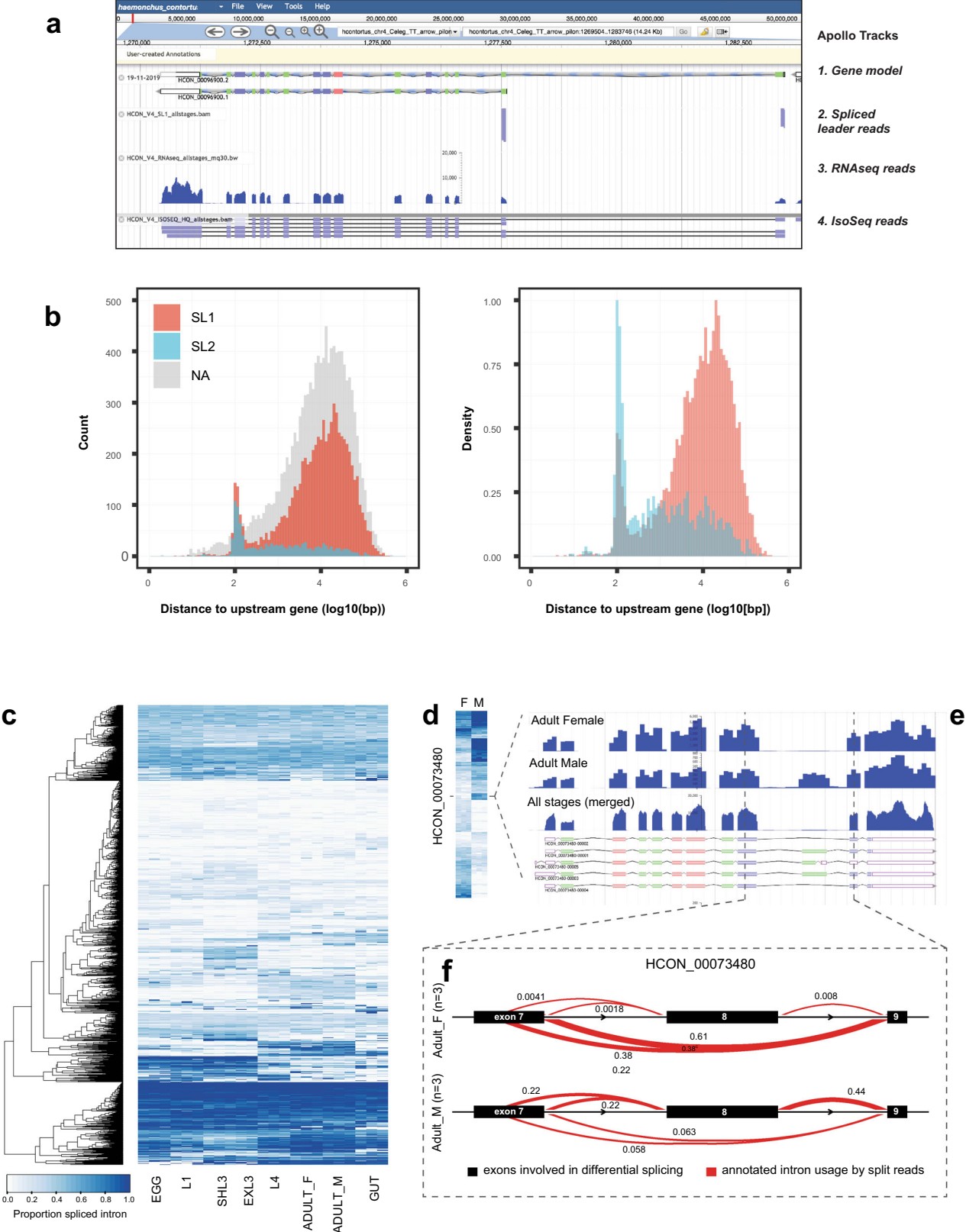

stages. We identify 1,055 genes with differentially spliced transcripts impacting at least one intron, representing on average 283 genes (FDR < 0.05; range: 43–418) that switch between different isoforms in each pairwise comparison of life stages throughout the lifecycle (Fig. 5c; Supplementary Table 7 [summary], Supplementary Data 8 [gene lists]). For example,

231 of 1,068 genes are differentially spliced between the adult male and female samples. The most differentially spliced intron is located in *HCON_00073480*, a one-to-one ortholog of *C. elegans daz-1* (Fig. 5d). DAZ-1 is an RNA-binding protein involved in oogenesis but not spermatogenesis in *C. elegans*[51]; in *H. contortus*, *HCON_00073480* is predominantly expressed

**Fig. 5 Extensive trans- and cis-splicing of gene transcripts throughout the life cycle. a** Example of a gene model, *HCON_00096900*, that contains two SL1 trans-spliced sites upstream of the transcriptional start sites of two distinct isoforms, each supported by full-length Iso-Seq sequence reads. **b** Distribution of intergenic distance (log10[bp]) between the start coordinates of genes containing SL1 (red), SL2 (blue) sequences, or with no detectable SL sequence (grey; NA), and the end coordinate of the gene immediately upstream. In the second plot, the relative frequency of SL1 or SL2 sequences is presented. **c** Heatmap of the top 100 differentially spliced introns identified in pairwise comparisons throughout the life cycle. Data are presented as the proportion of differentially excised intron, ranging from 0 (light) to 1 (dark), for each intron cluster for which differential splicing was identified. A complete list of genes with evidence of differential splicing is presented in Supplementary Data 7. **d** Heatmap of the top 100 differentially spliced introns between the adult male and adult female worms. The scale is the same as shown in (**c**). **e** RNA-seq expression data for the adult female, adult male, and all life stages combined, relative to the annotated isoforms for *HCON_00073480*, a one-to-one ortholog of the *C. elegans daz-1*. **f** Quantification of differential splicing of *HCON_00073480* between adult female (top) and adult male (bottom) samples, determined using leafcutter. Focused on exons 8, 9, & 10 (black bars), the red lines depict the linking of splice junctions, with the thickness of the line relative to the proportion of reads that support that junction. In this example, two differential splicing events are observed; (i) alternate splice donors of exon 8, and (ii) exon skipping of exon 9.

in females (average TPM = 636.22; highest expression in hermaphrodites in *C. elegans*), and is expressed in males, albeit approximately 23-fold less (average TPM = 26.71)(Fig. 5e). Only a single transcript is described in *C. elegans*, however, five isoforms are annotated in the *H. contortus* genome, with sex-specific expression of exon 9 that is present in males and absent in females. Further, differential splicing was found in the splice donor site of exon 8, and while both alternate splice sites are present in both sexes, the sites are used equally in males but the longer transcript is used in approximately 60% of transcripts in female worms (Fig. 5f).

**Distribution of global genetic diversity throughout the chromosomes.** *H. contortus* is a globally distributed parasite, characterised by large population sizes and in turn, high genetic diversity[27–29,52]. We have characterised the chromosomal distribution of genetic diversity by exploiting a recent large scale analysis of global genetic diversity[27] together with data from 74 additional worms from previously unsampled regions including Pakistan, Switzerland, USA, and the United Kingdom (Fig. 6a; https://microreact.org/project/hcontortus_global_diversity). Principal component analysis (PCA) of mtDNA diversity (Fig. 6b; Supplementary Fig. 9) showed that relationships between isolates are congruent with those previously described[27]; further, we show that Pakistani worms were genetically positioned between South African and Indonesian individuals, whereas the US and the UK samples were found among globally distributed samples that likely shared a common ancestral population that was distributed globally via modern human movement[27]. The US and UK samples consisted of both laboratory-maintained strains and field isolates; notably, apart from MHco3(ISE).N1, the laboratory strains retained similar levels of genetic diversity to the field samples. MHco3(ISE), from which the reference strain MHco3(ISE).N1 was derived, is believed to have originated in East Africa; although we have no samples from this region, the MHco3(ISE).N1 population was genetically similar to one of the three South African populations (Supplementary Fig. 10; Kokstad population), perhaps reflecting its African origins.

Genome-wide nucleotide diversity (π) is broadly consistent chromosome-wide between the autosomes (Fig. 6c–g). The high frequency of SNPs private to only a single population (relative to variants shared between two or more populations) represents a considerable proportion of the overall variation (23.55% of total variation), consistent with previous reports describing high regional genetic diversity[27–29]. On the X chromosome, nucleotide diversity is much lower than neutral expectation (i.e., $\pi_X/\pi_{autosome} \simeq 0.75$[53]) at a $\pi_X/\pi_{autosome}$ median ratio of 0.363 (Fig. 6h), which suggests a sex-ratio bias in favour of males. This observation, together with the distinct repeat patterns that were negatively correlated with recombination rate, suggests that recombination may be less frequent on the X chromosome than expected.

Interestingly, variation in X chromosome coverage (Supplementary Fig. 11a) increases in populations that are more genetically distinct from the reference strain (Supplementary Fig. 11b).

Finally, we reveal multiple discrete regions in each chromosome with high $F_{ST}$ among populations (Fig. 6c–h; bottom plots). Consistent with previous reports, strong genetic selection for anthelmintic resistance has impacted genome-wide variation; the most differentiated region is found on chromosome 1 surrounding the beta-tubulin isotype 1 locus (*HCON_00005260*; 7,027,492 to 7,031,447 bp), a target of benzimidazole-class anthelmintics for which known resistant mutations have been identified in these populations[27], as well as a broad region of elevated $F_{ST}$ from approximately 35 to 45 Mbp on chromosome 5 previously shown to be associated with ivermectin resistance in controlled genetic crosses[54]. The top 1% of $F_{ST}$ outlier regions per chromosome (280 10 kbp regions in total; Supplementary Data 9) contains 269 genes enriched for 10 GO terms (7 MF & 3 BP; https://biit.cs.ut.ee/gplink/l/1EEZzm6USa), largely describing proteolytic and cysteine-type peptidase activity.

## Discussion

The *H. contortus* MHco3(ISE).N1 V4 assembly presented represents the largest *de novo* chromosome-scale nematode genome to date and provides a critical genomic resource for a broad group of human and veterinary pathogens present among the Clade V nematodes. This assembly is complemented with a high-quality *de novo* genome annotation to provide a precise transcriptomic resource for within and between species comparison, improved UTR and isoform classification for functional genomics, and should help to improve the annotation of other nematode species. Finally, we provide a comprehensive description of genetic and transcriptomic diversity within individuals and across the species, which informs not only our understanding of key developmental trajectories but also the parasite's capacity to adapt when faced with challenges such as drug exposure.

A key challenge to achieving this chromosomal assembly was the presence of extensive genetic diversity in the sequencing data. Although high-quality genome assemblies are becoming more prevalent through the use of long-read sequencing technologies[3,4,55], these assemblies are typically derived from a single, few, or clonal individual(s) in which the genetic diversity is low or absent; most of the genetic diversity present in a single diploid individual can be phased and separated into distinct haplotypes[6,7], and assembled separately. However, even under these scenarios, heterozygosity remains a key technical limitation towards achieving contiguous genome assemblies. Considering the small physical size of *H. contortus*, it was necessary to pool thousands of individuals to generate sufficient material for long-read sequencing and while we did so from a semi-inbred population, there was still high genetic diversity present in the population of worms sequenced. The generation of such diversity

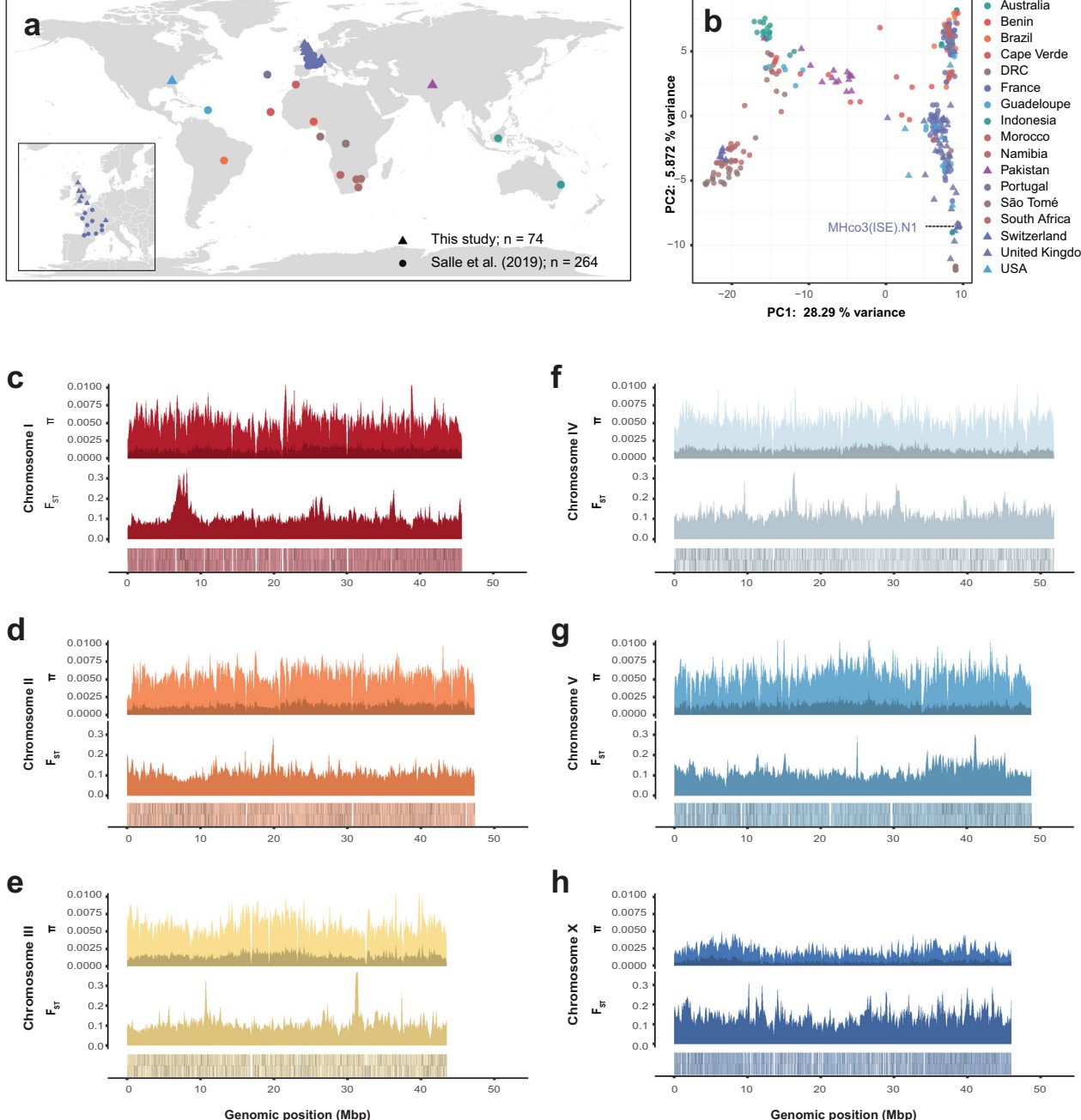

**Fig. 6 Genome- and population-wide genetic diversity. a** Global distribution of populations from which genomic data were derived, including 264 samples from Sallé et al. (circles) and 74 new samples (triangles) presented. The insert highlights the dense sampling from the United Kingdom (this study) and France[27]. **b** Principal component analysis of genetic diversity among the globally distributed samples, using 2,401 mtDNA variant sites. The reference strain, MHco3(ISE).N1, is indicated. **c–h** Chromosome-wide comparison of genetic diversity. Each comparison contains three panels. Top: Estimate of nucleotide diversity (π) calculated in 100 kbp windows, accounting variants found only in a single population (dark) or shared between two or more populations (light). Middle: Estimate of genetic differentiation ($F_{ST}$) between populations, measured in 100 kbp windows. Bottom: Gene density per chromosome, where each gene is represented is a black line on the sense (upper lines) or antisense (lower lines) DNA strand of the chromosome.

from a single mating pair is likely influenced by reproductive traits including polygamous mating and high fecundity of *H. contortus* female worms. How such haplotypic diversity is maintained in the population or is tolerated within single individuals remains unclear, however, it is likely that the holocentric chromosomes of *H. contortus* tolerate high genetic diversity during homologous pairing of chromosomes at meiosis[56]. Despite this diversity, we curated a major haplotype in the ISE.N1 population to represent the chromosomes and demonstrated how this haplotype and alternate haplotypes segregated in this semi-

inbred population. The absence in the literature to date of such chromosome-scale assemblies for physically small and genetically diverse eukaryotic species likely reflects the necessary investment required to improve a draft assembly to chromosomes. However, as DNA input requirements for long-read sequencing decrease[57], the ability and ease with which highly contiguous genomes can be generated for small and genetically diverse species will be greatly enhanced.

Completion of the X chromosome represents one of the key advancements in the current genome assembly. In doing so, we

revealed striking differences between the autosomes and X chromosome in (i) the repeat distribution and content, (ii) an enrichment for the longest genes of the genome, (iii) lower gene density and gene coverage, (iv) the lack of apparent genetic diversity, and (v) differences in the gene content that regulates X chromosome dosage and sex determination relative to *C. elegans*. Although we currently have little empirical data on recombination rates on the X chromosome, less recombination is expected on the X chromosome relative to the autosomes due to recombination on X only occurring during female gametogenesis. Similarly, the distinct repeat domain structure does suggest that recombination patterns may be different from the autosomes and warrants further investigation. The difference in nucleotide diversity between the X chromosome and autosomes further emphasises the impact of putative sex biases on the evolution of the X chromosome. *H. contortus* is polyandrous[31,35] and highly fecund, so that reproductive fitness is likely over-dispersed between males and females, potentially resulting in different effective population sizes for male and female parasites. These differences may be exacerbated if males and females are differentially affected by anthelmintic exposure as has been demonstrated for some compounds[58]. The dichotomy between the low nucleotide diversity and high variance between samples in the global cohort perhaps reflects a key limitation on the ability of a single linear reference genome to capture genome diversity in a species with high genetic diversity. Long-read sequencing data from *Haemonchus* populations and bioinformatics approaches designed to capture more complex variation should help reveal the full extent of genetic variation in this species and shed light on the role of structural variation in shaping this genetic variation.

Our high-quality annotation allowed us to extend previous pairwise transcriptomic comparisons and focus on the trajectories of genes with correlated expression throughout the life cycle. These data identified putatively co-regulated gene sets, providing greater insight into coordinated developmental programmes than has been described previously. The broad-scale coordinated gene expression associated with ion transport, channel activity, and transmembrane signalling receptor activity throughout the life-cycle was of interest, particularly as ion channels and receptors are the primary targets of several anthelmintic drugs used to control *H. contortus* and other helminth species of human and veterinary importance[59]. The downregulation of sensory and signalling pathway-related genes from pairwise comparisons of juvenile to adult stages has been described previously in both parasitic[24,60–63] and free-living[64,65] Clade V nematodes, which suggests this is a broadly conserved phenomenon throughout nematode development. However, a better understanding of these changes may explain why some parasites at particular life stages are less sensitive to these drugs[66,67], and may provide the rationale for selective treatment based on the known expression of drug targets. We further explored transcriptome variation by defining and quantifying differential isoform usage throughout the life cycle, a feature which is increasingly implicated in the sensitivity of drug targets in nematodes such as *H. contortus*; for example, sex-dependent sensitivity of emodepside is mediated by differential-splicing of the potassium channel *slo-1*[58], and similarly, expression of truncated acetylcholine receptor subunits *acr-8* and *unc-63* isoforms are associated with levamisole resistance[68,69]. Our results almost certainly under-estimate the extent of *cis*-splice variation within and between life stages; stringent filtering resulted in only a subset of genes tested (range: 759–1,156 genes per pairwise comparison of life stages), and yet, in each pairwise comparison, approximately 27.32% of genes on average contained evidence of splice variation (Supplementary Table 14). Further, each pairwise comparison identified cryptic

splice sites, indicating that additional novel isoforms undergoing differential splicing were missing from the genome annotation. Ongoing manual curation to the genome annotation to classify isoform variants, together with more fine-grained quantification of transcript usage, will provide additional insight into the novelty that differential isoform usage provides, particularly for those genes for which isoform switching, but not differential expression, occurs between key life-stage transitions. Beyond the life cycle, the impact of high genetic diversity on transcriptomic variation within or between wild isolates is not yet clear, although both technical and biological variation between three divergent strains has recently been described[70]. A greater understanding of this genetic-to-transcriptomic variation is required, especially if phenotypic traits associated with transcriptomic differences, e.g. increased gene expression of a drug transporter correlated with resistance, are to be correctly attributed to genetic selection.

Finally, the V4 assembly allowed the extent of genome rearrangements between *C. elegans* and *H. contortus* to be characterised, revealing numerous gene and repeat family expansions that were missed in the previous assembly. As further genomes are completed from the Strongylid clade, determining the rates of structural evolution within and between chromosomes in this important group of parasitic organisms should become possible. Investigating such evolutionary processes will be important for understanding genome-wide variation associated with trait variation, including adaptive traits such as response to climate change and anthelmintic resistance. *H. contortus* is an established model for understanding parasite response to anthelmintics and subsequent evolution of resistance to key drugs used to control parasitic nematodes[22]. Our chromosomal assembly has been instrumental in mapping genetic variation associated with ivermectin[54] and monepantel[71] resistance. In the case of ivermectin, the identification of the major genomic region on chromosome 5 associated with resistance would not have been possible without a chromosome-scale assembly and highlights the limitations of interpreting such variation using less-contiguous genome assemblies as a reference[30].

## Conclusions

Our chromosome-scale reference genome and comprehensive annotation provides a step forward toward resolving the complete genome composition and transcriptional complexity of *H. contortus*. The increasing availability of long-read sequencing together with advances in the sequencing of small diverse organisms holds great promise for the rapid generation of high-quality contiguous genomes for organisms such as *H. contortus*; future sequencing, including the generation of new genomes and transcriptomes *de novo* from individuals sampled throughout its geographic range, will provide great insight into how genetic variation is maintained and changes in response to persistent challenges, including anthelmintic exposure and climatic change, in this hyper-diverse globally important pathogen.

## Materials and methods

**Parasite material**. All parasite material used in the genome assembly was isolated from the semi-inbred strain of *H. contortus*, MHco3(ISE).N1. This strain was derived from a single round of single-worm mating using the MHco3(ISE) strain (protocol and procedure described in Sargison et al.[34]), and maintained as previously described[23].

**Iterative improvement of the genome assembly**. Our approach to improving the V1 genome to the final V4 chromosomal assembly is outlined in Supplementary Fig. 1. The sequence data generated for this study are described in Supplementary Data 1.

Initial manual improvement on the V1 genome focused on iterative scaffolding with *SSPACE*[72] and gap-filling with *IMAGE*[73] using Illumina 500 bp and 3 kbp libraries, with additional low-coverage data from 3, 8 and 20 kbp libraries generated using Roche 454 sequencing. These improvements were performed

alongside breaking of discordant joins using *Reapr*[74], and visual inspection using *Gap5*[75]. Substantial genetic variation was present in the sequence data due to the sequencing of DNA derived from a pool of individuals, resulting in a high frequency of haplotypes that assembled separately and therefore present as multiple copies of unique regions in the assembly. We surmised that much of the assembly fragmentation was due to the inability of scaffolding tools to deal with the changing rates of haplotypic variation, so we attempted to solve this manually in *Gap5*. As we had insufficient information to correctly phase these haplotypes, we chose, instead, the longest scaffold paths available accepting that our scaffolds would not represent single haplotypes but would rather be an amalgamation of haplotypes representing single chromosomal regions. This approach was initially difficult and time-consuming and was further confounded by a large number of repetitive sequences present in each haplotype.

Significant improvements in scaffold length were gained by the integration of OpGen (http://www.opgen.com/) optical mapping data. Optical mapping was performed following methods described previously[11] with the following exceptions: high molecular weight DNA was prepared by proteinase K lysis of pools of ~500 *H. contortus* L3 embedded in agarose plugs, after which one of three restriction enzymes (KpnI, AflII and KpnI) were used to generate three separate restriction map datasets. Initial attempts to generate a *de novo* assembly using optical molecules alone was unsuccessful, and therefore, optical contigs were generated using DNA sequence seeds from the genome assembly using *GenomeBuilder* (OpGen) and visualised and manually curated using *AssemblyViewer* (OpGen). Although this approach was successful, it was limited by the quality and integrity of the gap-dense scaffolds and arbitrary nature of the haplotype scaffolding.

Subsequent integration of PacBio long-read data alongside the optical mapping data resulted in major increases in contiguity. PacBio sequencing libraries were prepared and sequenced using the PacBio RSII system. A total of 32.3 Gbp raw subreads ($n = 4,085,541$, N50 = 10,299 bp) were generated from 33 flow cells, representing approximately 133.8× coverage (based on the estimated genome size of 283 Mbp). A *de novo* PacBio assembly was generated using *Sprai* (v0.9.9.18; http://zombie.cb.k.u-tokyo.ac.jp/sprai/index.html), which was mapped to the assembly and used to manually resolve many gaps and resolve some of the phasing between haplotypes. Using the *Sprai* PacBio *de novo* assembly, we were also able to incorporate contigs that were previously missing from the working assembly. The increase in contiguity of the PacBio assemblies, further improved using *canu* v1.3[76], revealed two major but diverse haplotype groups segregating at approximately 65 and 30% frequency in the pooled individuals sequenced; the presence of such diverse haplotypes resulted in a significantly expanded assembly over 500 Mbp. The major haplotype was more contiguous and therefore was chosen as the primary sequence to incorporate and improve the assembly. This approach was supported by competitive mapping of a single worm short read sequencing library (ENA: ERS086777), which was found to preferentially map to different scaffolds and/or different contigs within scaffolds that we inferred were different haplotypes present in this single worm. Regions in the chromosome with no ERS086777 coverage, but for which an alternate haplotype was identified with coverage from the PacBio assembly, were manually removed from the main chromosomes and replaced. Further, this selective mapping correlated well with the optical contigs, and once these sequences were removed, much better optical alignments were produced further improving the assembly. Alternate haplotypes from the PacBio assembly and those removed from the main assembly were analysed as a separate haplotype assembly.

The increase in contiguity and resolution of shorter repetitive regions using PacBio began to reveal chromosome-specific repetitive units. Although these repeats were highly collapsed in the assembly and were typically not spanned by optical molecules, we were able to iteratively identify and join contigs/scaffolds flanking large tandem repeats that had clear read-pair evidence that they only occurred in a single location in the genome (i.e., read pairs were all mapped locally once the join was made). These were further supported by local assemblies of subsets of PacBio reads that contained copies of the repeat regions followed by de novo assembly using *canu*[76] to reconstruct the flanking unique regions surrounding a repeat. This iterative process resulted in the production of chromosome-scale scaffolds, each terminating with a 6 bp repeat consistent with being a telomeric sequence (sequence motif: TTAGGC).

The V3 assembly, used in the *H. contortus* genetic map[35], consisted of five autosomal chromosomes with the X chromosome in two major scaffolds. The X chromosome-associated scaffolds were identified by synteny to the *C. elegans* X chromosome using *promer*[77] and by the expected ~0.75× coverage relative to the autosomes in pooled sequencing (due to a mixture of male and females present) and ~0.5× in single male sequencing libraries. We resolved the X chromosome using linked read long-range sequencing using the 10X Genomics Chromium platform (www.10xgenomics.com). DNA prepared from pooled worms was used to generate 10X sequencing libraries, which were subsequently sequenced on a HiSeq 2500 using 250 bp PE reads. Reads were mapped using *bwa mem* (v0.7.17-r1188), followed by scaffolding using *ARCS*[78] and *LINKS*[79]. A single scaffold was generated using this approach, merging the two major X-linked scaffolds as well as two short sequences putatively identified as being X-linked but previously unplaced (Supplementary Fig. 2). The completion of the X chromosome, together with the polishing of the genome first with *arrow* (https://github.com/PacificBiosciences/GenomicConsensus) followed by *Pilon* v1.22[80], finalised the genome to produce the V4 version presented here.

**Genome completeness**. Assembly completeness was determined throughout the improvement process using Core Eukaryotic Genes Mapping Approach (*CEGMA*; v2.5)[81] and Benchmarking Universal Single-Copy Orthologs (*BUSCO*; v3.0; –mode genome –lineage metazoa_odb9 –long)[82] gene completeness metrics. For comparison, these metrics were also determined for the complete set of nematode genomes from the WormBase ParaSite version 12 release.

**Haplotype analyses**. The identification and discrimination of the major and minor haplotypes in our assembly offered an opportunity to characterise the distribution of haplotypes throughout the genome, and the relative diversity between them. To visualise the broad-scale variation between the chromosome and haplotypes, we generated an approximate genome graph using *minigraph* (https://github.com/lh3/minigraph; parameters: -xggs). The output GFA file was visualised using *Bandage*[83]; a representative set of haplotypes were chosen for further comparison, and was compared using nucmer and visualised using *Genome Ribbon*[84]. To quantify the distribution and diversity of haplotypes, we used *minimap* to align haplotypes per chromosome. Haplotype density was calculated in 1 Mbp windows throughout the genome using *bedtools coverage*[85] and visualised using the R package *circlize*[86]. We determined the relative usage of detected haplotypes in the MHco3(ISE).N1 population by using Illumina sequencing of DNA from 11 individual worms and competitively mapping the sequencing data to both the reference and haplotype assemblies using *bwa mem*[87]. Relative sequence coverage was determined using *bedtools coverage*[85], which was normalised per sample to enable equivalent comparison between samples.

**Transcriptome library preparation and sequencing**. We made use of the RNA-seq data generated for each of the life stages as previously described[23] together with additional full-length cDNA (PacBio Iso-Seq) sequencing performed with the specific aim of generating evidence for the annotation. Three life stages were used in the Iso-Seq analyses: adult female, adult male, and L3. RNA was extracted by homogenising the samples in Trizol in Fast prep (MP bio) followed by chloroform extraction and ethanol precipitation of the RNA, after which 1.2 μg of total RNA was converted to cDNA using SmartSeq2 protocol[88]. Size fractionation of equimolar pooled cDNA of the different stages was performed using a SageELF electrophoresis system (Sage Sciences), from which cDNA size fraction pools spanning 400 to 6,500 bp were collected. To improve the coverage of transcripts that differed in abundance and also in length (and to help minimise library preparation and sequencing biases towards shorter and highly abundant transcripts), we used a cDNA normalisation method followed by size fractionation of the pooled normalised cDNA to recover discrete size fractions of transcripts. The cDNA normalisation, which exploits a duplex-specific nuclease (DSN), was performed using the Trimmer-2 cDNA normalisation kit (Evrogen) following the manufacturer's instructions. Size fractionated cDNA pools were sequenced on five individual flow cells on the RSII instrument. To increase the yield of the sequencing data, a subsequent sequencing run was performed after pooling the size fractions using a single flow cell on the Sequel instrument. A comparison of the sequences generated by the RSII and Sequel systems is presented in Supplementary Table 5.

**Generation of a high-quality non-redundant Iso-Seq dataset**. Full-length Iso-Seq subreads were mapped to the V4 genome using *minimap2* (version 2.6-r639-dirty[89]) with splice-aware mapping enabled (parameter: -ax splice). The generation of high-quality Iso-Seq reads was performed following the *Iso-Seq3* pipeline (https://github.com/PacificBiosciences/IsoSeq3), which consisted of four steps: (i) the generation of circular consensus sequencing (CCS) reads from raw subreads (ccs subreads.bam subreads.ccs.bam –noPolish –minPasses 1), (ii) the removal of library primers, and reorientation of reads (lima subreads.ccs.bam primers.fa subreads.demux.bam –isoseq –no-pbi), (iii) clustering of transcripts, removal of concatemers and trimming of polyA tails (isoseq3 cluster subreads.demux. primer_5p–primer_3p.bam subreads.isoseq3_unpolished.bam), and finally, (iv) polishing of the transcripts to generate a high quality set of isoforms (isoseq3 polish merged_subreads.isoseq3_unpolished.bam merged_subreads.bam merged_subreads.isoseq3_polished.bam).

**Gene model prediction and genome annotation**. A schematic of the strategy used for genome annotation is presented in Supplementary Fig. 6. Existing RNA-seq data were individually mapped to the V4 genome using *STAR* v2.5.2[90] before combined into a single bam using *samtools merge*. The merged bam was used as input to *Braker* (version 2.0)[37,91], which can exploit mapped RNA-seq information as hints to train the gene prediction tools *GeneMark-EX* and *Augustus*, substantially improving genome annotation. The use of exon, intron, and other hints derived from RNA-seq or proteins mapped to the genome can be used to train *Augustus* directly; initial attempts to incorporate mapped Iso-Seq transcripts as an input to *Braker*, or by training *Augustus* using exon and intron hints, were successful and produced superior gene models compared with *Braker* using RNA-seq alone. However, as Iso-Seq data encode full-length transcripts that include UTRs in a single sequence, and that in this data there was little discrimination between the depth of coverage between UTR and exon, many UTRs were incorrectly annotated as 5' or 3' coding exons. Accordingly, only short-read RNA-seq data were used as input to *Braker*, resulting in a primary annotation (GFF₁; Supplementary Fig. 6).

To incorporate Iso-Seq data into the genome annotation, we used the *PASA* (v2.2.0) genome annotation tool[92] to map the high-quality full-length transcripts to the genome using both *blat* and *gmap*, akin to the use of a *de novo* transcriptome assembly produced from, for example, *trinity*, to generate a transcript database (https://github.com/PASApipeline/PASApipeline/wiki/PASA_comprehensive_db). The *Braker* GFF$_1$ was subsequently loaded into *PASA* and integrated with the mapped Iso-Seq transcripts to produce the first round merged annotation (GFF$_2$; Supplementary Fig. 6).

To incorporate further coding evidence into the annotation, we used *EvidenceModeller* v1.1.1[38] to apply a differential weighting schema to multiple evidence sources before merging them into a single genome annotation. Evidence used included the GFF$_1$ and GFF$_2$ annotations from *Braker* and *PASA* as ab initio and transcript evidence, respectively, and protein sequences from *C. elegans* and *H. contortus* V1 genome assemblies (obtained from Wormbase ParaSite release 9 caenorhabditis_elegans.PRJNA13758.WBPS9.protein.fa.gz and haemonchus_contortus.PRJEB506.WBPS9.protein.fa.gz, respectively), as well as 110 curated genes; each protein dataset was mapped to the genome using *exonerate* v2.2.0 (parameters: –model protein2genome –percent 50 –showtargetgff[93]) followed by GFF extraction using *Exonerate_to_evm_gff3.pl* from *EvidenceModeller*. The weighting was applied as follows: PROTEIN (hc_v1 * curated models) = 2; PROTEIN (Ce) = 1; ABINITIO_PREDICTION (GFF$_1$ from Braker) = 2; TRANSCRIPT (GFF$_2$ from PASA) = 5. The merged output of *EvidenceModeller* (GFF$_3$) was imported back into *PASA* to update gene models with the full-length transcript information, including updates to UTRs and evidence for alternative splicing. Consistent with the author's recommendations, two rounds of iterative update and comparison were performed to maximise the use of the full-length transcript, i.e., Iso-Seq, data (GFF$_4$; Supplementary Fig. 6). Gene IDs were subsequently renamed with the prefix HCON and numbered with an 8 digit identifier that incremented by a value of 10 to allow identifiers to be subsequently added for new features if required (GFF$_5$; Supplementary Fig. 6).

The genome annotation, as well as available evidence used to generate the annotation (including per life stage RNA-seq, Iso-Seq [full-length high quality & CCS reads], coverage and repeat tracks), was uploaded to the web portal *Apollo*[94] for further visualisation and manual curation. This platform has been used to provide a means to continually improve the *H. contortus* genome annotation via manual curation as new information becomes available. These annotation improvements will be subsequently incorporated into WormBase ParaSite.

Annotation statistics were determined using *Genome Annotation Generator (GAG)*[95]. Annotation comparison was only made between the previously published V1 draft assembly[23], the final V4 chromosomal assembly presented here, the McMaster[24] and NZ isolate[33] draft assemblies, and the *C. elegans* reference; no comparable annotation was generated for the intermediate assemblies generated before V4 was finalised, including assembly versions V2 and V3. Annotation comparisons to a curated gene set to determine the sensitivity and specificity of the approaches used were performed using *gffcompare* v0.10.1 (http://ccb.jhu.edu/software/stringtie/gffcompare.shtml), and were further analysed for completeness using *BUSCO* (–mode proteins –lineage metazoa_odb9).

The repetitive content of the genome was analysed using *RepeatModeller* (v1.0.11) followed by *RepeatMasker* (v4.0.7). Further analysis of LTRs was performed using *LTRharvest* and *LTRdigest*[96] from the *GenomeTools* v1.5.9[97] package.

**Transcriptome analyses**. Quantitative transcriptome analyses were primarily performed using *kallisto* v0.43.1[98], which quantifies transcripts by pseudo-aligning raw sequencing reads to a transcriptome reference. The transcriptome reference was derived from the reference FASTA and annotation GFF3 data using *gffread* (https://github.com/gpertea/gffread). Raw sequencing reads from each of the life-stages were pseudo-aligned to the reference using *kallisto quant* (parameters: –bias –bootstrap-samples 100 –fusion), before the differential expression between pairwise combinations of life-stages was determined using *sleuth*[99]. For each transcript, differential expression between life-stages was tested using a Wald test, for which a q-value less than 0.05 was deemed significant.

To explore coordinated gene expression throughout the life-stages, we used *clust* v1.8.4[100] to automatically define and group transcripts by expression level profiles. Transcripts per million (TPM) counts per life-stage were generated from the kallisto analysis described above and used as input to *clust*. To explore putative regulatory sequences that may be shared by co-expressed genes, we searched in each group of clustered transcripts for conserved motifs in the 100 bp region immediately upstream of the start codon for each transcript using *DREME*[101], after which motif-containing sequences were identified using *FIMO*[102] (MEME v5.0.4). To perform gene set analyses, subsets of *H. contortus* genes together with associated *C. elegans* orthologs were obtained from WormBase ParaSite BioMart[17]. Functional enrichment analyses for both species were performed using *g:Profiler* (version e97_eg44_p13_d22abce), with a g:SCS multiple testing correction method applying a significance threshold of 0.05[103]. *OrthoFinder* v2.2.7[104] was used to determine orthologous relationships between protein sequences inferred from the *H. contortus* V1 and V4, the McMaster *H. contortus*, as well as *H. placei* and *C. elegans* protein-coding gene annotations.

**Trans-splicing: characterisation of splice leader usage and distribution**. We characterised the number and distribution of genes for which mapped RNA-seq transcripts contained the canonical *H. contortus* SL1 (GGTTTAATTACC-CAAGTTTGAG) and SL2 (GGTTTTAACCCAGTATCTCAAG) sequences. For each splice leader sequence, *cutadapt* v1.13[105] was used to detect (allowing up to 10% divergence in sequence match [–error-rate = 0.1] and minimum match length of 10 bp [–overlap=10]) and subsequently trim the leader sequences from raw RNA-seq reads from each of the life stages, after which all trimmed reads were collated and mapped to the genome using *hisat2* v2.1.0[106]. The relative coverage of the sequence between the trimmed splice leader site and (i) the start codon (using a window 200 bp upstream and 50 bp downstream of the start codon) and (ii) internal exons (excluding the 1st exon), was determined using *bedtools coverage*. *Weblogo*[107] was used to visualise sequence conservation in the genomic sequence surrounding the splice leader site, and in the splice leader sequences themselves. The spliced leader sequence splice sites for all life stages were combined into a single dataset for plotting and quantitative analyses.

**Cis-splicing: differential isoform usage using Leafcutter**. To analyse the extent of differential spliced isoform usage between life-stages, we used *leafcutter*[108](see http://davidaknowles.github.io/leafcutter/ for installation and scripts described below) to quantify the proportion of reads that span introns and then subsequently quantify differential intron usage across samples. First, raw RNA-seq reads were mapped to the reference genome using *STAR*[90] (parameters: –twopassMode basic –outSAMstrandField intronMotif –outSAMtype BAM Unsorted) to generate BAM files per life-stage. Intron junctions (junc files) were generated from the BAM files, which were subsequently clustered using *leafcutter_cluster.py* (parameters: -m 30 -l 500000). Differential introns between pairwise life-stages using *leafcutter_ds.R* (parameters: –min_samples_per_intron=3 –min_samples_per_group=3 –min_coverage=10), which required the clustered introns and an exon file (derived from the annotation GFF3) as input. Preparation of the data for visualisation of differentially spliced introns was performed using *prepare_results.R;* an annotation GTF was required as input, which was generated from the annotation GFF3 using *gffread*. The visualisation was performed using *run_leafviz.R*.

**Population genetic analyses**. To understand how genetic variation was distributed throughout the chromosomes, we collated sequencing data from our recent global analysis[27] together with in-house archival sequencing data to form the most complete sample cohort of *H. contortus* whole-genome data to date. Reanalysis of the global cohort data was necessary, as the previous analysis was performed using the V3 assembly. In total, 338 samples from 46 populations in 17 countries were analysed; the samples together with geolocation, mitochondrial phylogeny describing genetic connectivity between samples, and ENA references to individual sequencing datasets are available to explore using *Microreact*[109] and can be visualised here: https://microreact.org/project/hcontortus_global_diversity.

Raw sequencing data were mapped to the V4 reference using *bwa mem*[87] (parameters: -Y and -M options were used to use soft clipping for supplementary alignments, and to mark shorter hits as secondary, respectively), followed by identification of duplicate reads using *Picard* (v2.5.0; https://github.com/broadinstitute/picard). Samples for which multiple sequencing lanes were generated were mapped independently and subsequently combined into a single dataset using *samtools merge*. Mapping statistics were generated using *samtools* (v1.3) *flagstat* and *bamtools* (v 2.3.0) *stats* and compared using *MultiQC* v1.3[110].

Genetic variation within and between samples was determined using *GATK* (v 4.0.3.0). GVCFs were first created using *HaplotypeCaller*; to improve efficiency, this process was performed in parallel for each chromosome for each sample separately, and subsequently combined by chromosome using *CombineGVCFs*. Variants were determined per chromosome using *GenotypeGVCFs*, after which the final raw variants were combined using *vcftools concat*. A hard filter was applied using *GATK VariantFiltration* to mark variants using the following criteria: QD < 2.0, FS > 60.0, MQ < 40.0, MQRankSum < −12.5, and ReadPosRankSum < −8.0. Nucleotide diversity and *F*-stats within and between populations was determined using *vcftools* v0.1.14[111] --window-pi and --weir-fst-pop, respectively, using a window size of 100 kbp.

To analyse mitochondrial DNA diversity, bcftools view was used to select single nucleotide variants with a read depth > 10, homozygous variant, and with minor allele frequency >0.01. To generate the phylogeny, variants were used to generate a consensus mitochondrial genome per sample, which were aligned using *mafft* v7.205[112], and from which maximum likelihood phylogenies were generated using *iqtree* v1.6.5[113] (parameters: -alrt 1000 -bb 1000 -nt 1) using automated ModelFinding[114]. Principal component analysis (PCA) of the mitochondrial DNA variation was performed using the *poppr*[115] package in R.

Sequencing coverage variation between the autosomes and X chromosome for each BAM file was performed using *samtools*[116] *bedcov*, using a BED file of 100 kbp windows generated using *bedtools*[85] *makewindows*.

**Statistics and reproducibility**. All statistical analyses, data exploration, and visualisation were performed using *R* (v3.5.0; https://www.R-project.org/) unless otherwise stated.

**Ethics approval**. The use of experimental animals to maintain parasite populations for the purposes described in this manuscript was approved by the Moredun Research Institute Experiments and Ethics Committee and were conducted under approved British Home Office licenses in accordance with the Animals (Scientific Procedures) Act of 1986. The Home Office licence numbers were PPL 60/03899 and 60/4421, and the experimental identifiers for these studies were E06/58, E06/75, E09/36 and E14/30.

**Reporting summary**. Further information on research design is available in the Nature Research Reporting Summary linked to this article.

## Code availability

Code and workflows used to analyse data and reproduce figures are available at https://github.com/stephenrdoyle/hcontortus_genome and Zenodo https://doi.org/10.5281/zenodo.4069270.

## Data availability

The raw sequence data is available under the ENA accession PRJEB506, with reference to specific sequencing libraries described throughout the text, and/or in Table S3 of Laing et al. 2013. RNA-seq data is available from the ENA study ID PRJEB1360. The genome assembly has been made available at ENA (assembly accession: GCA_000469685.2) and WormBase ParaSite (https://parasite.wormbase.org/Haemonchus_contortus_prjeb506/Info/Index). A static version of the genome annotation used in this paper is available at ftp://ftp.sanger.ac.uk/pub/pathogens/sd21/HCON_V4_GENOME/ (signoff date: 25th Jan 2019), however, the most up-to-date version of the annotation can be accessed at WormBase ParaSite (https://parasite.wormbase.org/Haemonchus_contortus_prjeb506/Info/Index/[17]).

Genome variation data can be visualised using MicroReact (https://microreact.org/project/hcontortus_global_diversity), from which links to ENA accessions for individual sample sequencing data from the global collection can be obtained. All relevant data are available from the corresponding authors upon request.

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

## Acknowledgements
We would like to acknowledge members of the BUG Consortium and Parasite Genomics group at the Wellcome Sanger Institute for insightful discussions over the course of this project. We also thank Pathogen Informatics and DNA Pipelines (WSI) for their support and expertise, and the Biosciences Division at the Moredun Research Institute for expert care and assistance with animals. Work described here was supported by Wellcome's core funding of the Wellcome Sanger Institute (grant WT206194), a Wellcome Trust Project Grant to JSG (grant 067811), and the Biotechnology and Biological Sciences Research Council (BBSRC) grants (BB/M003949/1, BB/P024610/1 and BB/K020048/1). JDN was supported by a Graduate Mobility Award from McGill University.

## Author contributions
J.A.C., J.S.G., M.B., E.D. and S.R.D designed the study. S.R.D. led the analysis and interpretation of the data, A.T., A.M., H.B., K.B., R.B. and J.D.N., analysed data. W.B., K.L.H, M.P. and F.H.R., provided novel software and computational resources. R.L., D.B., N.S., U.C., M.Z.S, J.W., E.D., C.B. and G.Sal., maintained parasites, collected and/or provided samples. G.San., M.A.Q., E.R. and K.M., performed laboratory work. N.H. coordinated samples and sequencing. S.R.D. and J.A.C. wrote the manuscript, with contribution and review of the final manuscript by all authors.

## Competing interests
The authors declare no competing interests.

## Additional information

Stephen R. Doyle [1][✉], Alan Tracey [1], Roz Laing [2], Nancy Holroyd[1], David Bartley[3], Wojtek Bazant[1], Helen Beasley[1], Robin Beech[4], Collette Britton[2], Karen Brooks[1], Umer Chaudhry[5], Kirsty Maitland[2], Axel Martinelli[1], Jennifer D. Noonan [4], Michael Paulini[6], Michael A. Quail[1], Elizabeth Redman[7], Faye H. Rodgers [1], Guillaume Sallé[8], Muhammad Zubair Shabbir[9], Geetha Sankaranarayanan[1], Janneke Wit [7], Kevin L. Howe[6], Neil Sargison[5], Eileen Devaney [2], Matthew Berriman [1], John S. Gilleard [7] & James A. Cotton[1][✉]

[1]Wellcome Sanger Institute, Hinxton, Cambridgeshire CB10 1SA, UK. [2]Institute of Biodiversity Animal Health and Comparative Medicine, College of Medical, Veterinary and Life Sciences, University of Glasgow, Garscube Campus Glasgow G61 1QH, UK. [3]Moredun Research Institute, Pentlands Science Park, Bush Loan Penicuik EH26 0PZ, UK. [4]Institute of Parasitology, McGill University, 21111 Lakeshore Road, Sainte Anne-de-Bellevue, QC H9X3V9, Canada. [5]Royal (Dick) School of Veterinary Studies, University of Edinburgh, Edinburgh EH25 9RG, UK. [6]European Molecular Biology Laboratory, European Bioinformatics Institute, Hinxton, Cambridgeshire CB10 1SA, UK. [7]Department of Comparative Biology and Experimental Medicine, Faculty of Veterinary Medicine, University of Calgary, Calgary, AB, Canada. [8]INRAE - U. Tours, UMR 1282 ISP Infectiologie et Santé Publique, Centre de recherche Val de Loire, Nouzilly, France. [9]University of Veterinary and Animal Sciences, Lahore 54600, Pakistan. [✉]email: stephen.doyle@sanger.ac.uk; jc17@sanger.ac.uk

