## [Peer Review File · Communications Biology]

Reviewers' Comments:

Reviewer #1:

Remarks to the Author:

GENERAL COMMENTS

Doyle et al. present the assembly and biological analysis of their chromosomal-quality genome sequence for the blood-feeding parasitic nematode *Haemonchus contortus*. Having arisen from *Haemonchus* ancestors which parasitized wild antelopes in sub-Saharan Africa, *H. contortus* has spread to infect (and sometimes kill) goats and sheep worldwide, becoming the world's most economically important parasite of farm animals. As a member of the strongylid class of parasitic nematodes, *H. contortus* is closely related to many other gastrointestinal parasites of sheep, cows, and other domesticated animals, as well as to hookworms that infect over 400 million people. Because of its importance, *H. contortus* was the first strongylid nematode for which genomes and transcriptomes were assembled and analyzed: two parallel genomes with accompanying RNA-seq data were published in 2013.

Seven years later, the new *H. contortus* genome assembly described here by Doyle et al. was badly needed. Short-read Illumina sequencing technology used for the 2013 assemblies yielded much lower genomic contiguity and completeness than what Doyle et al. have been able to achieve here, using third-generation sequencing technology (long reads and chromosomal scaffolding). Their new genome assembly, unlike its predecessors, has both complete representation of gene content and chromosomal contiguity. Among other benefits, these features allow reliable genetic mapping of population-specific variations (which shed light on ecology and evolution of *H. contortus*), as well as reliable genetic mapping of biologically important traits such as drug resistance.

Because of the exceptional genetic diversity of *H. contortus*, even in a laboratory strain that had been inbred for years, assembling *H. contortus* to this quality was not a trivial matter of getting long reads and then just turning an assembly program's crank. Extensive work was required to identify and sort out haplotypes so that a single canonical genome sequence could be defined. This took unusual effort by Doyle et al., and their success are impressive.

Doyle et al. have biologically characterized their chromosomal *H. contortus* genome assembly in several ways: chromosome-wide features such as repetitive DNA types; comparisons of gene synteny and chromosomal organization to the model nematode *Caenorhabditis elegans*; predictions of protein-coding gene structure and observations of operon structure (with chained mRNAs and trans-spliced 5' mRNA leaders) based on both short-read and long-read RNA-seq data; identification of 19 developmentally coregulated gene sets, along with their statistically enriched biological functions and associated non-coding DNA motifs (with possible cis-regulatory activity); analysis of differential splicing during development; and extensive analysis of genetic diversity from *H. contortus* populations worldwide. The scope of these analyses is (unfortunately) quite seldom seen for any nematode genome other than *C. elegans*: for instance, I am not aware of any genomewide analysis of developmentally variable exon splicing in any nematode other than *C. elegans*, even though we know from study of *C. elegans* itself that such developmentally regulated exon-switching is likely to be developmentally important.

These analyses synergistically improve one another. Their analysis of developmental gene expression is enhanced by improved gene predictions, previously published RNA-seq data, and newly available software for automatic clustering of gene expression modules. Likewise, their analyses of microsynteny between *H. contortus* and *C. elegans* gain biological meaning from their identification of operons in *H. contortus* (the latter having not yet been achieved for any other parasitic nematode, to my knowledge). Their analysis of global genetic diversity is enabled by both improved gene predictions and the chromosomal contiguity of their genome (allowing chromosomal domains with varying levels of diversity to be identified).

The text of the manuscript is clearly and cogently written, making it both informative and enjoyable to read. The supplementary tables, a utilitarian but essential component of serious genome papers, are extensive and well-organized, beginning with a highly useful summary for all figures and tables. Pleasingly, the many different gene sets listed in supplementary data tables (sharing developmental patterns of gene expression or exon splicing) are hyperlinked to archived g:Profiler analyses that allow viewing of enriched Gene Ontology function terms for each gene set, which in turn should allow other researchers to explore hypotheses raised by these gene sets. The methods are notably well-documented, with full identification of programs used with their version numbers; this is an important point for genome papers, given how much of their content relies on how computational analyses were done in precise detail. Their data transparency is excellent. All of their raw sequence data have already been submitted to major public databases (ENA/NCBI). Although their genome assembly and gene annotations do not seem to have become available yet in ENA/NCBI, Doyle et al. generously made their genome publicly available in the ParaSite database (parasite.wormbase.org) well before submitting this manuscript for publication, and have encouraged other researchers to use their pre-publication genome and its gene predictions without any restrictions. The figures are well-done, and make the mostly text-based data of this genome paper easier to absorb. Citations of past literature are appropriate and useful.

With this work by Doyle et al., past genomic analyses of *H. contortus* have become historical relics. I predict that their new *H. contortus* genome will become a key resource for understanding and control of strongylid parasites in humans and other mammals, and that their initial analysis of that genome will become a model towards which other parasitic nematode genome projects will aspire.

The overall quality of this work leaves me with only the following to add: a request for uploading genome annotations to ENA/NCBI; a scientific comment on comparative transcriptomics of clade V nematodes; and some minor corrections and comments.

REQUEST FOR UPLOADING TO ENA/NCBI

As noted above, the authors have done a superb job of making their genomic data publicly available. I found only one oversight, which for all I know the authors are in the process of correcting as I write this review. Although their final genome assembly (V4) and its gene predictions are publicly available in the ParaSite database (parasite.wormbase.org), they are *not* (as far as I can tell from checking with BlastP on April 4, 2020) yet publicly available on ENA/NCBI. Although ParaSite is much better than nothing, it is not nearly as good as having the V4 genome and its gene predictions also archived in ENA/NCBI, since many biological researchers will not be aware of ParaSite, or may find it inconvenient to add ParaSite to their genomic analyses. Moreover, many large-scale genomic databases only work with protein-coding genes that have passed through ENA/NCBI and been further archived in RefSeq or UniProt.

I thus strongly encourage the authors to upload their V4 genome assembly and its protein-coding gene annotations to ENA/NCBI, if they have not already done so; I also encourage them to make this upload publicly visible, if V4 and its gene annotations have already been uploaded but are still being kept private.

COMMENT ON COMPARATIVE TRANSCRIPTOMICS

Among their many other discoveries, Doyle et al. note on page 15:

"Functional characterisation of the cluster 1 genes [genes with higher expression in free-living stages that subsequently decreased in expression in parasitic stages] revealed 45 significantly enriched gene ontology (GO) terms (26 molecular function [MF], 13 biological process [BP], & 6 cellular compartment (CC) terms; <https://biit.cs.ut.ee/gplink/I/8jEckGwQQS>) that predominantly

described ion transport and channel activity, as well as transmembrane signalling receptor activity."

Doyle et al. later comment on page 28:

"The broad-scale coordinated gene expression associated with ion transport, channel activity, and transmembrane signalling receptor activity throughout the lifecycle was of interest, particularly as ion channels and receptors are the primary targets of a number of anthelmintic drugs used to control *H. contortus* and other helminth species of human and veterinary importance [ref. 59] . A better understanding of these changes may explain why some parasites at particular life stages are less sensitive to these drugs [refs. 60,61] , and may provide the rationale for selective treatment based on the known expression of drug targets."

This is indeed an important finding, and I agree with Doyle et al. that it has notable implications for anthelmintic drug discovery. However, this is not the very first time that this pattern of gene expression has been observed in a strongylid nematode; moreover, this pattern of gene expression is not confined to parasitic nematodes, but is also found in the related free-living nematode *C. elegans*.

In their genomic and transcriptomic analysis of the hookworm *Ancylostoma ceylanicum*, Schwarz et al. ([2015], *Nat. Genet.* 47, 416-422, PubMed 25730766) noted:

"G protein-coupled receptors (GPCRs), receptor-gated ion channels and neurotransmission-related functions in general were downregulated [in *A. ceylanicum*] during early infection (L3i to 24.PI), along with transcription factors (Supplementary Tables 9b and 11b). We observed the same pattern among genes downregulated in the transition from L3 to fourth-stage (L4) larvae both in *H. contortus* [ref. 21] and *C. elegans* [ref. 42] (Supplementary Table 8). This finding is consistent with downregulation after L3 of sensory perception and transcription genes in both *C. elegans* [ref. 43] and *N. americanus* [ref. 24] and of ion channel genes in *A. caninum* and *Brugia malayi* [refs. 32,44]. Such downregulation might thus be conserved in both parasitic and free-living nematodes."

Downregulation of *H. contortus* genes implicated in ion receptor and transmembrane receptor activity remains an important finding of this paper. However, Doyle et al. may wish to put their finding in a larger context by citing past work which shows this pattern of gene expression to be found in both other parasitic *and* non-parasitic clade V nematodes.

MINOR CORRECTIONS AND COMMENTS

Page 2:

"were reported in 2013, however, both were" would probably read better as "were reported in 2013; however, both were".

"only a single group of 10 orthologs are colinear between the genomes of the two species": would it be possible for the authors to add a list of these 10 orthologous colinear genes (with both *H. contortus* and *C. elegans* gene names) to their supplementary data tables? It is interesting that such a 10-gene cluster exists, and it would be good for readers of this article to have some way of finding out what it is! For one thing, they could then check to see whether this same 10-gene cluster is also found in other clade V nematode species; if it were, this would suggest that some remarkable selective force is conserving their organization to an exceptional degree.

More generally, could this be extended to giving lists of the top 10 blocks of gene colinearity? As indicated in Supplementary Figure S3D, there is only one block containing 10 syntenic genes, but there are nine more blocks that contain 9 syntenic genes apiece. The authors should not be

required to give gene lists ad nauseam, but the most extensive conserved syntenic blocks are either freaks of randomness or evolutionarily noteworthy.

Page 3:

"The assembly is approximately 283.4 Mbp in length" should read "The assembly is 283.4 Mbp in length".

Page 4:

The phylogenetic analysis of BUSCO orthology (Figure S4 A), showing that specific losses of BUSCO genes have phylogenetically intelligible patterns (and thus implying strongly that the losses are biologically real) is a nice touch.

Page 12:

"low-abundant transcripts" should read "low-abundance transcripts".

Page 13 (and elsewhere):

"RNAseq" should be spelled "RNA-seq" both here and throughout the paper.

Page 13:

"the close relationship between *H. contortus* and *H. placei*, a gastrointestinal pathogen commonly associated with cattle." This statement is true but not referenced; for readers who are not already soaked in the *Haemonchus* literature, a well-chosen reference would be good here. It is up to the authors which one they would like to use, but one possibility would be to cite Chaudhry et al. ([2015], *Int. J. Parasitol.* 45, 149-159, PubMed 25449043). This reference is one of the more recent and more heavily cited articles showing that *H. contortus* and *H. placei* are closely related enough species to allow rare hybridization; it also contains references to earlier (pre-2015) literature on *H. placei*.

Page 15:

"lowly expressed": I am aware that this wording is used somewhat often in modern genomics, but I consider it an unfortunate neologism. The authors might consider revising the relevant sentence from this:

"Consistent with the pattern of highly variable genes, two dominant clusters of genes were identified: (i) genes with higher expression in free-living stages that subsequently decreased in expression in parasitic stages (cluster 1; n = 1,550 genes; Figure 4 B) and (ii) genes that were lowly expressed in free-living stages that increased in expression in parasitic stages (cluster 8; n = 1,542 genes)."

to this:

"Consistent with the pattern of highly variable genes, two dominant clusters of genes were identified: (i) genes with higher expression during free-living stages that subsequently decreased in expression during parasitic stages (cluster 1; n = 1,550 genes; Figure 4 B) and (ii) genes with lower expression during free-living stages that subsequently increased in expression during parasitic stages (cluster 8; n = 1,542 genes)."

Page 15:

"cysteine-type peptidase activity in cluster 8 (GATAAGR ...)": this observation is striking for two reasons. As Doyle et al. are well aware, intestinal upregulation of cysteine-type peptidase genes has long been recognized to be a key trait of strongylid parasitic nematodes, and gut expression of such peptidases is considered essential for successful parasitism. Furthermore, the motif described here strikingly matches the AHTGATAARR motif bound by the ELT-2 intestinal transcription factor in *C. elegans* (McGhee et al. [2007], *Dev. Biol.* 302, 627-645, PubMed 17113066; McGhee et al. [2009], *Dev. Biol.* 327, 551-565, PubMed 19111532). This similarity suggests that strongylid parasites might have evolved strong upregulation of intestinal cysteine proteases by coopting gene expression modules that already existed in their free-living ancestors (which are likely to have resembled the free-living clade V nematodes *C. elegans* and *Pristionchus pacificus*).

I do not make this comment in order to suggest any revisions of the current paper. But I think this particular finding by Doyle et al. points towards future possible analyses of the evolution of parasitism.

Page 16:

"...absent, including *xol-1* and *sdc-1*, -2, and -3.": This passage is confusing and raises troubling inconsistencies between Figure 4 and Supplementary Table S11.

Given the results shown in Figure 4, it should instead read: "...absent, including *xol-1* and *sea-1*, -2, and -3." In Figure 4, *sdc-1*, -2, and -3 are shown as present in *H. contortus*.

However, when I go on to look at the more detailed results listed in Supplementary Table S11, I see that not only *sea-1*/*-2*/*-3* are missing from *H. contortus*, but that *sdc-1*/*-2*/*-3* are missing as well! Seeing this inconsistency motivated me to check for other inconsistencies: other possible inconsistencies seem to include *dpy-26*, *gld-3*, *tra-2*, and *nos-3* (all of which are shown as present in *H. contortus* in Figure 4, but are listed as absent from *H. contortus* in Supplementary Table S11).

The authors should either correct these inconsistencies (if they are real) in Figure 4 and Supplementary Table S11, or explain to me what it is that I have obtusely failed to understand (if the inconsistencies are only apparent) before publishing this analysis.

Page 19:

"...a small proportion of SL1-spliced genes are found less than 100 bp downstream of the nearest gene." Further down, the authors note that "variations have been described in *C. elegans*" and cite a review about *C. elegans* operons by Blumenthal et al. (WormBook 2015 Apr 28, PubMed 25936768).

In that review, Blumenthal et al. notes that "... hybrid operons contain a promoter at the 5' end of the cluster and a different promoter somewhere within the cluster. Presumably, the internal promoter results in expression of the gene or genes downstream of it at a time or place where the 5' end promoter is not expressed, or is expressed at an insufficiently high level. The internal promoter allows differential expression of the genes in the operon in response to particular signals, for example, but still allows the entire cluster to be expressed as a unit when needed. Both co-regulation and differential expression are thereby achieved.", and cites published work showing that some *C. elegans* operons contain secondary internal promoters (and are thus "hybrid operons"; Huang et al. [2007], *Genome Res.* 17, 1478-1485, PubMed 17712020).

Thus, it seems to me that the observation of internal SL1 splices in *H. contortus* operons might suggest that some hybrid *H. contortus* operons, like those in *C. elegans*, contain secondary internal promoters.

Page 24:

"The top 1% of F ST outlier regions per chromosome (280 10 kbp regions in total) contains 268 genes enriched for 10 GO terms (7 MF & 3 BP; <https://biit.cs.ut.ee/gplink/l/HKi5aHzqTa>), largely describing proteolytic and cysteine-type peptidase activity. Analysis of d N /d S from alignments of 9,970 one-to-one orthologs from H. contortus and the closely related H. placei identified 108 genes putatively under positive selection (d N /d S > 1)... We extended this analysis to H. contortus variation derived from the global diversity dataset; after filtering, 849 (4.8% of 17,607) genes were identified with a d N /d S greater than 2."

Could Doyle et al. please add three Supplementary Tables that list these sets of 268, 108, and 849 genes, respectively? All three sets are potentially of considerable interest to other researchers analysing parasitic nematode genomes; but, as things stand (with no supplementary gene lists available), it will be needlessly laborious and time-consuming for those other researchers to have to independently compute what those gene sets actually contain.

Note: I am aware that one URL given by Doyle et al. (<https://biit.cs.ut.ee/gplink/l/HKi5aHzqTa>) leads to a g:Profiler page that includes a list of the 268 genes. However, this will not necessarily be obvious to the readers, and it does not provide the permanent availability that adding a supplementary table listing the 268 genes will have. Moreover, there is no corresponding Web link for the sets of 108 or 849 genes at all.

Page 24:

"...largely describing proteolytic and cysteine-type peptidase activity. Analysis of dN/dS from alignments of 9,970 one-to-one orthologs..." This passage might be more readable if the authors began a new paragraph at the word "Analysis..."

"We did, however, identify a number of genes likely associated with host/parasite interactions...": this might be clearer if it were written as "Among the set of 108 genes, however, we did identify a number of genes that are likely associated with host/parasite interactions..."

"...were among the most abundant pfam domains." This comes after a list of orthologs of C. elegans genes, i.e., after a list of specific *genes* rather than a list of protein domains. Perhaps Doyle et al. could better rewrite this as "...were among the most abundant categories of gene types." or something of that sort. Moreover, even if they wish to stick with the current wording, "pfam domains" should be written "Pfam domains" (i.e. with a capital 'P' in 'Pfam').

Page 36:

"As such, only short-read RNAseq data were used..." might be better written as "Accordingly, only short-read RNA-seq data were used..." (or "Therefore...").

Page 38:

"For each transcript, differential expression between life-stages was tested using a Wald test..." This raises an issue with analyzing gene expression with kallisto. The default mode for kallisto analysis is to distinguish expression between *transcripts*, not *genes*; and these are not the same (a gene can have two or more transcripts through alternative splicing). Since sleuth was designed to work with the outputs of kallisto, this default mode of kallisto operation affects sleuth as well. However, the supplementary data tables make it clear that Doyle et al. mapped the kallisto and sleuth results to H. contortus genes at some point (in particular, Supplementary Table S8 shows kallisto/sleuth results for transcripts, but then somehow these results were mapped to genes). How was this done? The mapping may have been simple, but I would still like the authors to explain and clarify this point.

Page 41:

"[dN/dS] ratios greater than one indicate positive selection for new protein-coding changes, ratios close to zero indicate neutral evolution, and ratios less than one indicate purifying selection." Mostly this makes sense, but the part about 'ratios close to zero indicate neutral evolution' strikes me as being badly wrong; I think it should read 'ratios close to *one* indicate neutral evolution'. If I am wrong, can the authors please explain why? If I am right, can the authors please correct this error?

Legend of Supplementary Figure S1:

"we present the changes in the assembly overtime" should read "we present the changes in the assembly over time".

Figure S4:

In some sense, the division of syntenic gene pairs into the categories of "conserved order" and "conserved order but reorientated" must be arbitrary; as the authors indicate in the main text, although the six chromosomes of *H. contortus* are clearly homologous, they preserve no long-range synteny whatsoever (as shown in Figure 1B). The initial decision about which way to orient a given chromosome (e.g., chromosome I) of *H. contortus* with respect to *C. elegans* therefore must have been done at random. That being said, the nearly perfect split between fully conserved versus conserved-but-reorientated syntenic pairs (Supplementary Figure S3C) is what one would expect given extensive shuffling of microsyntenic genomic regions.

Legend of Supplementary Figure S7:

"transcripts with significant but shared differential expression": Although I share the authors' admiration for the clust program and use it for my own work, it does not provide statistical estimates (such as p-values) for the nonrandomness of the expression clusters which it generates. I would thus reword this to something like "transcripts with shared differential expression".

"putative co-regulated transcripts" should read "putatively co-regulated transcripts".

Figure S9: This is excellent! It is the first time I have seen multipopulation PCA diversity maps presented in a way that makes it easy for the reader to identify each population in the map (by giving many different maps, one for each distinct population, in which that one population is marked while all other data points are kept constant).

Supplementary Table S2:

Data columns B and C are both labeled "*H. contortus* V4", which is redundant and confusing. I believe the authors may have meant "*H. contortus* V4 Chromosomes" (column B) and "*H. contortus* V4 Haplotypes" (column C), which would make much more sense, and which would be consistent with data column names used in Supplementary Table S3. Please review these column names and (if possible) correct them so that they are nonredundant and nonconfusing.

Supplementary Table S5:

The "Precision" values for gene models from the authors' previously published V1 genome assembly, in this table, look astoundingly low. They are in the low single digits (or even low *single decimals*) for every precision value: base, 2.7; exon, 2.6; intron, 2.7; transcript level, 0.7 [!]; and locus level, 0.8 [!]. In contrast, the precision values for all of the V4 gene models are much more sane (ranging from 29.8 to 98, with most values above 60). Are the V1 precision

values really that bad, or is this some sort of systematic typographical error?

Supplementary Table S6:

Data column C is labeled simply "H. contortus", which is confusing. I believe the authors may have meant "H. contortus V1" (i.e., their 2013 genome assembly which was the precursor for their V4 assembly now). Please review this column name and (if possible) correct it so that it is nonconfusing.

Supplementary Table S8:

"Threshold for DE was $q_value < 0.01$ and $b < -1$ or $b > 2$ " should read "Threshold for DE was $q_value < 0.01$ and $b < -1$ or $b > 2$ " (i.e., correct the spelling of 'Threshold').

Reviewer #2:

Remarks to the Author:

I find this a really good manuscript overall. The analysis is largely based on a good quality de novo genome assembly/annotation and the data and analyses are extremely high quality - in many ways I think this will be used as a model for many future papers on a diverse range of species.

I was initially slightly skeptical (before having read the full supplement) that the results might have been affected by potential issues derived from the fact that the genomes were not all generated in a single go. However, it's clear from the supplement that the authors have taken considerable steps to account for this, and thus I have no technical concerns at all.

I have only few minor comments:

1) CEGMA

CEGMA is discontinued, there is no need to use this, unless there is a specific feasible justification not reported here.

The CEGMA's authors, strongly recommend do not use this tool anymore: "We suggest you consider using alternative tools". The last update was in July 30th 2014.

<http://korflab.ucdavis.edu/datasets/cegma/>.

As I have mentioned before, this paper has an enormous potential to be used as a model for future studies. I truly believe CEGMA analyses should be removed in this paper.

I suggest to keep only the information from BUSCO.

2) Synteny

Pag 7: "Complete within-chromosome synteny would be expected to show a positive linear relationship between the genomic positions of orthologs between *C. elegans* and *H. contortus* chromosomes; "

"We show a remarkable pattern of almost complete conservation of chromosome content (synteny) with *Caenorhabditis elegans*, but almost no conservation of gene order."

Here, I really missed a reference to the statements. A reference to the evolutionary distance between both species and why higher conservation in the gene order was expected. We know that theoretically two orthologs genes should share homologous neighboring genes, however, there is a chance of non-homology matches occurring in collinearity.

Probably the authors are trying to tell a different history from the previous paper:<https://www.ncbi.nlm.nih.gov/pmc/articles/PMC4054779/pdf/gb-2013-14-8-r88.pdf>, but they cannot ignore the referential for the evolution. Also, looking for this previous work I would not expect a higher gene order conservation between *C. elegans* and *Haemonchus contortus*. Also, how can Trans and Cis splicing can affect the gene order, as it has been extensively described in *C. elegans*.

3) Figure 2. Haplotype and repeat distribution within and between chromosomes

That's a really bad legend. Most part of the text should be explained in the main text. Also, the ChrX (A,B) is quite confuse. The authors need to add one more letter (A,B,C,D) to explain this part.

Same for the Figure5. The legend contains a lot of unnecessary information.

4) Pag12: Generation of a high-quality transcriptome annotation incorporating short and long reads

This is an elegant strategy.

Could you please check the correct Apollo reference (Ref: 37)? I believe that you have used the WebApollo version (page: 37), if this is the case, please correct the reference.

WebApollo:

<https://genomebiology.biomedcentral.com/articles/10.1186/gb-2013-14-8-r93>

5) Figure S6. Annotation pipeline schematic used to incorporate RNAseq and IsoSeq data into a single annotation

The picture is in a low resolution. My reading was somewhat impaired.

6) RATT

Figure S6. Initial attempts to transfer genome annotations from the V1 assembly to the intermediate V3 assembly using tools such as Ratt [87] were unsuccessful, and with the generation of new Iso-Seq data, a new strategy was designed and implemented.

It's unclear to me (on page36) what exactly was done with the RATT. If it was just an unsuccessful transfer attempt, I believe this should not be mentioned.

The authors may decide to remove this information.

7) GAG

Was this tool used only to generate the annotation metrics? If so, it is ok!

If this tool has been used for some type of correction of the annotation, I am very distressed about the possible results, it may have implications for their quality of the annotation. This is a really bad and dangerous tool that unfortunately has been used a lot, but it shouldn't!

If the authors have another tool to reach the metrics, I strongly recommend not to mention the use of GAG.

8) Genome Annotation – Manual Curation

I remain in doubt: Was a manual curation of the genome generated?

If it was done, it is necessary to add a line paragraph of how it was generated.

No more comments.

Reviewer #3:

Remarks to the Author:

The manuscript by Doyle et al. presents a chromosome-scale assembly of the parasitic nematode *Haemonchus contortus*. This represents a major advancement over previously published genomes which

suffered from the high levels of genetic diversity in the sequenced population that caused overestimated genome sizes, substantial misassembly, and false signals of duplications.

The haplotype resolved assembly together with new Iso-seq and RNA-seq data allowed the authors to generate a vastly improved set of gene annotations and to use this new annotation to reinvestigate the developmental transcriptome and trans- and cis-splicing in this parasitic nematode. Furthermore, they provide an update on the population genomics of this species using newly sequenced isolates. Altogether, this manuscript represents a massive and highly important amount of work that definitely deserves publication.

I only have some minor comments.

abstract:

"The *H. contortus* MHco3(ISE).N1 genome assembly presented here represents the most contiguous and resolved nematode assembly outside of the *Caenorhabditis* genus to date," -> What about *Strongyloides ratti* and *Pristionchus pacificus*? I would prefer a more modest term such as "represents one of the most contiguous". Also, even if the statement is true, for how long will it hold?

p. 3

"smallest of genomes or for select organisms, where genome assembly projects"
-> "selected organisms" ?

p. 3

"the excessive diversity presents a significant challenge to the assembly process, and consequently, typically results in fragmented assemblies that contain misassembled and haplotypic sequences."

I think Barriere et al. *Genome Res.* 2009 Mar;19(3):470-80. is an important work that deserves to be cited here.

p. 6

I do not like the term "recombined gene pairs" because the link to the biological process of recombination is very loose. I would prefer something like "rearranged", "reshuffled".

Legend Figure 1

a) "Complete within-chromosome synteny would be expected to show a positive linear relationship between the genomic positions of orthologs between *C. elegans* and *H. contortus* chromosomes;"
-> "Complete within-chromosome collinearity" ?

b) Check that species names are in italic.

c) The lower density of genes in the center of the X chromosome is so striking that it should be

commented on.

d) Given that there are around 1000 dots on each of the plots, I am not sure if one can visually detect a positive linear relationship even if it existed. I wonder, how these plots would look like with the `smoothScatter` function in R that displays the density of data points.

p. 8

"high average orthologs per partial and complete CEGMA genes"
What does "average orthologs" mean?

p. 8

" When compared against publically available genomes in WormBase ParaSite, the V4 genome assembly represents the most contiguous and resolved nematode assembly outside of the *Caenorhabditis* genus, and second-most complete Clade V parasitic nematode assembly based on the presence of universally conserved orthologs ($n = 20$; *Ancylostoma ceylanicum* is the most complete based on BUSCO analysis; Additional file 1: Figure S4 A,B).
"  First, contiguity is usually measured in N50 (for example the *P. pacificus* assembly has a contig N50 of 4.6 p.7 (Rödelsperger et al. 2017), vs *Haemonchus* 3.8 Mbp p.7), Additional file 1: Figure S4 A,B only shows BUSCO data. Second, it is not obvious that *Haemonchus* is the most complete genome among the tested ones in Figure S4 A,B.

Fig 2C

a) this panel is far too small, I would recommend to split this figure into two.
b) how does repeat count (number of repeats) compare to repetitive fraction (Amount of sequence that is covered by repeats). In terms of gene density, people often use different terms which can easily cause misunderstanding.

p. 13

It would be nice to compare how the improved annotation is reflected at the BUSCO level.

Fig 3b Why is the feature count so important? Would not just showing the length distributions of genomic features in different *haemonchus* annotations be most informative?

p 18

"and sequence conservation in the genome sequence"
avoid to use "sequence" twice?

Fig 5B does NA refer to cases truly without splice leaders or ambiguous genes that are partially transcribed to SL1 and SL2? Are there examples of that?

Fig 5D females and males should be labeled.

Fig 5E What is the source of the gene models below the RNA-seq tracks

p 19.

"Although these variations have been described in *C. elegans* [47] , they seem to be more of the norm rather than the exception in *H. contortus*."

I think Sinha et al. RNA. 2014 Sep;20(9):1386-97. found similar patterns. This reference could be taken to support that this observation might not simply an exception in a weird parasite.

p 24

"We extended this analysis to *H. contortus* variation derived from the global diversity dataset; after filtering, 849 (4.8% of 17,607) genes were identified with a dN/dS greater than 2." $dN/dS > 2$ sounds very suspicious. This could be caused by misestimations in cases of very low overall divergence or very short proteins, probably also other causes like misalignments. Anyway, this screen for adaptation is highly speculative and in my opinion distracts from the main selling point of the paper (see general comments).

General comments:

Some sections such as the Gene Ontology analysis of expression clusters are very descriptive. Given that the manuscript is pretty long anyway, I would appreciate if the authors could stronger focus on the key findings of every analysis. For example, the population genomic analysis should stronger focus on patterns that could not have been characterized with previous versions of the *H. contortus* assembly and the new population data from Pakistan. I do not see why the UK vs French comparison is of importance here (p.22).

A lot of space in Figures is taken up by examples. I am not sure to what extent these examples are needed. These arrangements should be reconsidered.

Figure captions can be streamlined by adding proper legends to individual panels. e.g Fig 4c, Fig 2C,D

The authors compare version V1 and V4 of the *Haemonchus* annotations, a comment on V2 and V3 would be much appreciated.

The authors should avoid citing manuscripts that have not gone through peer review, e.g. biorxiv. Biorxiv is a nice platform to put ideas and to provide early access to new results, but these articles should not be cited if their results have not at least been approved by reviewers.

For transparency reasons, I would like to state that I received the invitation to review this manuscript on March 16th, accepted on March 20th, and submitted my review on March 20th

best regards,

Christian Rödelsperger

Reviewers' comments:

Reviewer #1 (Remarks to the Author):

GENERAL COMMENTS

Doyle et al. present the assembly and biological analysis of their chromosomal-quality genome sequence for the blood-feeding parasitic nematode *Haemonchus contortus*. Having arisen from *Haemonchus* ancestors which parasitized wild antelopes in sub-Saharan Africa, *H. contortus* has spread to infect (and sometimes kill) goats and sheep worldwide, becoming the world's most economically important parasite of farm animals. As a member of the strongylid class of parasitic nematodes, *H. contortus* is closely related to many other gastrointestinal parasites of sheep, cows, and other domesticated animals, as well as to hookworms that infect over 400 million people. Because of its importance, *H. contortus* was the first strongylid nematode for which genomes and transcriptomes were assembled and analyzed: two parallel genomes with accompanying RNA-seq data were published in 2013.

Seven years later, the new *H. contortus* genome assembly described here by Doyle et al. was badly needed. Short-read Illumina sequencing technology used for the 2013 assemblies yielded much lower genomic contiguity and completeness than what Doyle et al. have been able to achieve here, using third-generation sequencing technology (long reads and chromosomal scaffolding). Their new genome assembly, unlike its predecessors, has both complete representation of gene content and chromosomal contiguity. Among other benefits, these features allow reliable genetic mapping of population-specific variations (which shed light on ecology and evolution of *H. contortus*), as well as reliable genetic mapping of biologically important traits such as drug resistance.

Because of the exceptional genetic diversity of *H. contortus*, even in a laboratory strain that had been inbred for years, assembling *H. contortus* to this quality was not a trivial matter of getting long reads and then just turning an assembly program's crank. Extensive work was required to identify and sort out haplotypes so that a single canonical genome sequence could be defined. This took unusual effort by Doyle et al., and their success are impressive.

Doyle et al. have biologically characterized their chromosomal *H. contortus* genome assembly in several ways: chromosome-wide features such as repetitive DNA types; comparisons of gene synteny and chromosomal organization to the model nematode *Caenorhabditis elegans*; predictions of protein-coding gene structure and observations of operon structure (with chained mRNAs and trans-spliced 5' mRNA leaders) based on both short-read and long-read RNA-seq data; identification of 19 developmentally coregulated

gene sets, along with their statistically enriched biological functions and associated non-coding DNA motifs (with possible cis-regulatory activity); analysis of differential splicing during development; and extensive analysis of genetic diversity from *H. contortus* populations worldwide. The scope of these analyses is (unfortunately) quite seldom seen for any nematode genome other than *C. elegans*: for instance, I am not aware of any genomewide analysis of developmentally variable exon splicing in any nematode other than *C. elegans*, even though we know from study of *C. elegans* itself that such developmentally regulated exon-switching is likely to be developmentally important.

These analyses synergistically improve one another. Their analysis of developmental gene expression is enhanced by improved gene predictions, previously published RNA-seq data, and newly available software for automatic clustering of gene expression modules. Likewise, their analyses of microsynteny between *H. contortus* and *C. elegans* gain biological meaning from their identification of operons in *H. contortus* (the latter having not yet been achieved for any other parasitic nematode, to my knowledge). Their analysis of global genetic diversity is enabled by both improved gene predictions and the chromosomal contiguity of their genome (allowing chromosomal domains with varying levels of diversity to be identified).

The text of the manuscript is clearly and cogently written, making it both informative and enjoyable to read. The supplementary tables, a utilitarian but essential component of serious genome papers, are extensive and well-organized, beginning with a highly useful summary for all figures and tables. Pleasingly, the many different gene sets listed in supplementary data tables (sharing developmental patterns of gene expression or exon splicing) are hyperlinked to archived g:Profiler analyses that allow viewing of enriched Gene Ontology function terms for each gene set, which in turn should allow other researchers to explore hypotheses raised by these gene sets. The methods are notably well-documented, with full identification of programs used with their version numbers; this is an important point for genome papers, given how much of their content relies on how computational analyses were done in precise detail. Their data transparency is excellent. All of their raw sequence data have already been submitted to major public databases (ENA/NCBI). Although their genome assembly and gene annotations do not seem to have become available yet in ENA/NCBI, Doyle et al. generously made their genome publicly available in the ParaSite database (parasite.wormbase.org) well before submitting this manuscript for publication, and have encouraged other researchers to use their pre-publication genome and its gene predictions without any restrictions. The figures are well-done, and make the mostly text-based data of this genome paper easier to absorb. Citations of past literature are appropriate and useful.

With this work by Doyle et al., past genomic analyses of *H. contortus* have become historical relics. I predict that their new *H. contortus* genome will become a key resource for

understanding and control of strongylid parasites in humans and other mammals, and that their initial analysis of that genome will become a model towards which other parasitic nematode genome projects will aspire.

Response to review comment:

We thank the reviewer for their very positive and comprehensive appraisal of our work. We have tried to be as transparent and descriptive of the data and work performed as we could have done, in the hope that our resources generated and analyses presented will have as broad an impact as suggested.

The overall quality of this work leaves me with only the following to add: a request for uploading genome annotations to ENA/NCBI; a scientific comment on comparative transcriptomics of clade V nematodes; and some minor corrections and comments.

REQUEST FOR UPLOADING TO ENA/NCBI

As noted above, the authors have done a superb job of making their genomic data publicly available. I found only one oversight, which for all I know the authors are in the process of correcting as I write this review. Although their final genome assembly (V4) and its gene predictions are publicly available in the ParaSite database (parasite.wormbase.org), they are *not* (as far as I can tell from checking with BlastP on April 4, 2020) yet publicly available on ENA/NCBI. Although ParaSite is much better than nothing, it is not nearly as good as having the V4 genome and its gene predictions also archived in ENA/NCBI, since many biological researchers will not be aware of ParaSite, or may find it inconvenient to add ParaSite to their genomic analyses. Moreover, many large-scale genomic databases only work with protein-coding genes that have passed through ENA/NCBI and been further archived in RefSeq or UniProt.

I thus strongly encourage the authors to upload their V4 genome assembly and its protein-coding gene annotations to ENA/NCBI, if they have not already done so; I also encourage them to make this upload publicly visible, if V4 and its gene annotations have already been uploaded but are still being kept private.

Response to review comment:

We are happy that the reviewer has appreciated our attempts to make our data publicly available, and completely agree with the reviewer's comment that the genome and annotation should be made available via ENA/NCBI.

When we completed the genome assembly (version V4), we submitted it to ENA and it is available here: https://www.ebi.ac.uk/ena/data/view/GCA_000469685.2. However, the annotation was completed later, and due to some initial problems getting the genome uploaded at ENA, we had not followed up on submitting the annotation to ENA before we submitted our manuscript for publication. We are fully committed to propagating the genome update through to the ENA but there isn't a well-developed mechanism for doing that, especially as INSDC does not provide a mechanism for keeping the annotation updated as it is improved by curators (we are doing this via manual curation in Apollo). For this reason, our primary focus has been the public release through a different deposition mechanism, via WormBase Parasite and UniProt, that will ensure widespread availability of the latest gene models and curated protein set.

To address this comment, we have:

- Maintained a static version of the annotation on our FTP as described here: ftp://ftp.sanger.ac.uk/pub/pathogens/sd21/HCON_V4_GENOME/
- Communicated with UniProt, who have initiated the transfer of *H. contortus* proteins from WormBase Parasite to UniProt. This will occur very soon. Further, any updates to the annotation in WormBase Parasite will automatically be transferred to UniProt.
- In parallel, we will initiate submission of the annotation to ENA, however, this will take somewhat longer to appear than the previous two mechanisms.

COMMENT ON COMPARATIVE TRANSCRIPTOMICS

Among their many other discoveries, Doyle et al. note on page 15:

"Functional characterisation of the cluster 1 genes [genes with higher expression in free-living stages that subsequently decreased in expression in parasitic stages] revealed 45 significantly enriched gene ontology (GO) terms (26 molecular function [MF], 13 biological process [BP], & 6 cellular compartment (CC) terms; <https://biit.cs.ut.ee/gplink/l/8jEckGwQQS> [biit.cs.ut.ee]) that predominantly described ion transport and channel activity, as well as transmembrane signalling receptor activity."

Doyle et al. later comment on page 28:

"The broad-scale coordinated gene expression associated with ion transport, channel activity, and transmembrane signalling receptor activity throughout the lifecycle was of interest, particularly as ion channels and receptors are the primary targets of a number of anthelmintic drugs used to control *H. contortus* and other helminth species of human and veterinary importance [ref. 59] . A better understanding of these changes may explain why

some parasites at particular life stages are less sensitive to these drugs [refs. 60,61] , and may provide the rationale for selective treatment based on the known expression of drug targets."

This is indeed an important finding, and I agree with Doyle et al. that it has notable implications for anthelmintic drug discovery. However, this is not the very first time that this pattern of gene expression has been observed in a strongylid nematode; moreover, this pattern of gene expression is not confined to parasitic nematodes, but is also found in the related free-living nematode *C. elegans*.

In their genomic and transcriptomic analysis of the hookworm *Ancylostoma ceylanicum*, Schwarz et al. ([2015], *Nat. Genet.* 47, 416-422, PubMed 25730766) noted:

"G protein-coupled receptors (GPCRs), receptor-gated ion channels and neurotransmission-related functions in general were downregulated [in *A. ceylanicum*] during early infection (L3i to 24.PI), along with transcription factors (Supplementary Tables 9b and 11b). We observed the same pattern among genes downregulated in the transition from L3 to fourth-stage (L4) larvae both in *H. contortus* [ref. 21] and *C. elegans* [ref. 42] (Supplementary Table 8). This finding is consistent with downregulation after L3 of sensory perception and transcription genes in both *C. elegans* [ref. 43] and *N. americanus* [ref. 24] and of ion channel genes in *A. caninum* and *Brugia malayi* [refs. 32,44]. Such downregulation might thus be conserved in both parasitic and free-living nematodes."

Downregulation of *H. contortus* genes implicated in ion receptor and transmembrane receptor activity remains an important finding of this paper. However, Doyle et al. may wish to put their finding in a larger context by citing past work which shows this pattern of gene expression to be found in both other parasitic *and* non-parasitic clade V nematodes.

Response to review comment:

We thank the reviewer for reminding us of this previous literature and agree we have not given enough due credit to this previous work.

To address this comment, we have amended the discussion to state the following:

"The broad-scale coordinated gene expression associated with ion transport, channel activity, and transmembrane signalling receptor activity throughout the lifecycle was of interest, particularly as ion channels and receptors are the primary targets of several anthelmintic drugs used to control *H. contortus* and other helminth species of human and veterinary importance (Wolstenholme, 2011). The downregulation of sensory and signalling pathway-related genes from pairwise comparisons of juvenile to adult stages have been described previously in both parasitic (Schwarz et al. 2015; Schwarz et al. 2013; Tang et al.

2014; Wang et al. 2010; Choi et al. 2011) and the free-living (Gerstein et al. 2010; Kim et al. 2013) Clade V nematodes, which suggests this is a broadly conserved phenomenon throughout nematode development. However, a better understanding of these changes may explain why some parasites at particular life stages are less sensitive to these drugs (Basáñez et al., 2008; Sabah et al., 1986), and may provide the rationale for selective treatment based on the known expression of drug targets.“

MINOR CORRECTIONS AND COMMENTS

Page 2:

"were reported in 2013, however, both were" would probably read better as "were reported in 2013; however, both were".

Response to review comment:

Agreed and corrected as suggested.

"only a single group of 10 orthologs are colinear between the genomes of the two species": would it be possible for the authors to add a list of these 10 orthologous colinear genes (with both *H. contortus* and *C. elegans* gene names) to their supplementary data tables? It is interesting that such a 10-gene cluster exists, and it would be good for readers of this article to have some way of finding out what it is! For one thing, they could then check to see whether this same 10-gene cluster is also found in other clade V nematode species; if it were, this would suggest that some remarkable selective force is conserving their organization to an exceptional degree.

More generally, could this be extended to giving lists of the top 10 blocks of gene colinearity? As indicated in Supplementary Figure S3D, there is only one block containing 10 syntenic genes, but there are nine more blocks that contain 9 syntenic genes apiece. The authors should not be required to give gene lists ad nauseam, but the most extensive conserved syntenic blocks are either freaks of randomness or evolutionarily noteworthy.

Response to review comment:

We agree with the reviewer that we could and should have provided some more details on the specific genes in co-linear and potentially conserved blocks.

To address this comment we have provided the gene lists for colinear blocks with 5 genes or more in the new Supplementary Table S2, and have made a new panel within Figure S3 (E) showing the genome-wide distribution of colinear blocks with 5 or more genes.

Page 3:

"The assembly is approximately 283.4 Mbp in length" should read "The assembly is 283.4 Mbp in length".

Response to review comment:

Agreed and corrected as suggested.

Page 4:

The phylogenetic analysis of BUSCO orthology (Figure S4 A), showing that specific losses of BUSCO genes have phylogenetically intelligible patterns (and thus implying strongly that the losses are biologically real) is a nice touch.

Response to review comment:

We thank the reviewer for recognising this. Anecdotally, we and others have observed BUSCO often performs suboptimally for nematode and flatworm genomes, and so we sought to explain this missingness. Clearly, an improvement could be made to the training HMMs (certainly not within the scope of the work presented) to improve BUSCO detection of these groups of organisms.

Page 12:

"low-abundant transcripts" should read "low-abundance transcripts".

Response to review comment:

Agreed and corrected as suggested..

Page 13 (and elsewhere):

"RNAseq" should be spelled "RNA-seq" both here and throughout the paper.

Response to review comment:

Agreed and corrected as suggested. We also updated Isoseq to Iso-Seq to be consistent.

Page 13:

"the close relationship between *H. contortus* and *H. placei*, a gastrointestinal pathogen commonly associated with cattle." This statement is true but not referenced; for readers who are not already soaked in the *Haemonchus* literature, a well-chosen reference would be good here. It is up to the authors which one they would like to use, but one possibility would be to cite Chaudhry et al. ([2015], *Int. J. Parasitol.* 45, 149-159, PubMed 25449043). This reference is one of the more recent and more heavily cited articles showing that *H. contortus* and *H. placei* are closely related enough species to allow rare hybridization; it also contains references to earlier (pre-2015) literature on *H. placei*.

Response to review comment:

We thank the reviewer for highlighting this and agree that Chaudhry et al 2015 is a sensible reference to cite here. Now corrected.

Page 15:'

"lowly expressed": I am aware that this wording is used somewhat often in modern genomics, but I consider it an unfortunate neologism. The authors might consider revising the relevant sentence from this:

"Consistent with the pattern of highly variable genes, two dominant clusters of genes were identified: (i) genes with higher expression in free-living stages that subsequently decreased in expression in parasitic stages (cluster 1; n = 1,550 genes; Figure 4 B) and (ii) genes that were lowly expressed in free-living stages that increased in expression in parasitic stages (cluster 8; n = 1,542 genes)."

to this:

"Consistent with the pattern of highly variable genes, two dominant clusters of genes were identified: (i) genes with higher expression during free-living stages that subsequently decreased in expression during parasitic stages (cluster 1; n = 1,550 genes; Figure 4 B) and (ii) genes with lower expression during free-living stages that subsequently increased in expression during parasitic stages (cluster 8; n = 1,542 genes)."

Response to review comment:

We thank the reviewer for the suggested revision and have amended the text accordingly.

Page 15:

"cysteine-type peptidase activity in cluster 8 (GATAAGR ...)": this observation is striking for two reasons. As Doyle et al. are well aware, intestinal upregulation of cysteine-type peptidase genes has long been recognized to be a key trait of strongylid parasitic

nematodes, and gut expression of such peptidases is considered essential for successful parasitism. Furthermore, the motif described here strikingly matches the AHTGATAARR motif bound by the ELT-2 intestinal transcription factor in *C. elegans* (McGhee et al. [2007], *Dev. Biol.* 302, 627-645, PubMed 17113066; McGhee et al. [2009], *Dev. Biol.* 327, 551-565, PubMed 19111532). This similarity suggests that strongylid parasites might have evolved strong upregulation of intestinal cysteine proteases by coopting gene expression modules that already existed in their free-living ancestors (which are likely to have resembled the free-living clade V nematodes *C. elegans* and *Pristionchus pacificus*).

I do not make this comment in order to suggest any revisions of the current paper. But I think this particular finding by Doyle et al. points towards future possible analyses of the evolution of parasitism.

Response to review comment:

We thank the reviewer for making this observation and connection between our data and the *C. elegans* literature. This is a really interesting hypothesis that deserves some thought towards testing in future work.

Page 16:

"...absent, including *xol-1* and *sdc-1*, -2, and -3.": This passage is confusing and raises troubling inconsistencies between Figure 4 and Supplementary Table S11.

Given the results shown in Figure 4, it should instead read: "...absent, including *xol-1* and *sea-1*, -2, and -3." In Figure 4, *sdc-1*, -2, and -3 are shown as present in *H. contortus*.

However, when I go on to look at the more detailed results listed in Supplementary Table S11, I see that not only *sea-1*/*-2*/*-3* are missing from *H. contortus*, but that *sdc-1*/*-2*/*-3* are missing as well! Seeing this inconsistency motivated me to check for other inconsistencies: other possible inconsistencies seem to include *dpy-26*, *gld-3*, *tra-2*, and *nos-3* (all of which are shown as present in *H. contortus* in Figure 4, but are listed as absent from *H. contortus* in Supplementary Table S11).

The authors should either correct these inconsistencies (if they are real) in Figure 4 and Supplementary Table S11, or explain to me what it is that I have obtusely failed to understand (if the inconsistencies are only apparent) before publishing this analysis.

Response to review comment:

We appreciate the reviewer's effort in cross-checking our Figure 4 and Table S11, and indeed, the reviewer is correct in pointing out some inconsistencies. We apologise and acknowledge that these were an oversight.

In response to this comment, we have confirmed the data presented in Table S11, have updated Figure 4 and corrected the text to read as follows:

"In *C. elegans*, chromosome dosage is determined by measuring the expression of key genes on the X and autosomes; in *H. contortus*, only two of the *C. elegans* X-linked orthologs - *fox-1* and *sex-1* - are present, however, these are located on the autosomes, whereas the autosomal dosage associated genes *sea-1*, *-2*, and *-3*, are missing. Five of the six *C. elegans* orthologs associated with the dosage compensation complex (DCC) were found (*dpy-26* is missing), however, the genes *xol-1* and *sdc-1*, *-2*, and *-3* that act to initiate the recruitment of the DCC are absent."

Page 19:

"...a small proportion of SL1-spliced genes are found less than 100 bp downstream of the nearest gene." Further down, the authors note that "variations have been described in *C. elegans*" and cite a review about *C. elegans* operons by Blumenthal et al. (WormBook 2015 Apr 28, PubMed 25936768).

In that review, Blumenthal et al. notes that "... hybrid operons contain a promoter at the 5' end of the cluster and a different promoter somewhere within the cluster. Presumably, the internal promoter results in expression of the gene or genes downstream of it at a time or place where the 5' end promoter is not expressed, or is expressed at an insufficiently high level. The internal promoter allows differential expression of the genes in the operon in response to particular signals, for example, but still allows the entire cluster to be expressed as a unit when needed. Both co-regulation and differential expression are thereby achieved.", and cites published work showing that some *C. elegans* operons contain secondary internal promoters (and are thus "hybrid operons"; Huang et al. [2007], Genome Res. 17, 1478-1485, PubMed 17712020).

Thus, it seems to me that the observation of internal SL1 splices in *H. contortus* operons might suggest that some hybrid *H. contortus* operons, like those in *C. elegans*, contain secondary internal promoters.

Response to review comment:

We thank the reviewer for making this observation and agree that this seems like a sensible hypothesis for our observation.

In response to this comment, we have updated the text to read as follows:

“These observations suggest that while broadly consistent with the mechanism by which SL1/SL2 sequences are used in *C. elegans*, (i) *H. contortus* SL1-sequences can be found in downstream genes within a polycistron, and may reflect evidence of hybrid operons that contain a second, internal promoter to control the expression of a subset of genes within the operon (Huang et al. 2007),...”

Page 24:

"The top 1% of F_{ST} outlier regions per chromosome (280 10 kbp regions in total) contains 268 genes enriched for 10 GO terms (7 MF & 3 BP; <https://biit.cs.ut.ee/gplink/l/HKI5aHzqTa> [biit.cs.ut.ee]), largely describing proteolytic and cysteine-type peptidase activity. Analysis of dN/dS from alignments of 9,970 one-to-one orthologs from *H. contortus* and the closely related *H. placei* identified 108 genes putatively under positive selection (dN/dS > 1)... We extended this analysis to *H. contortus* variation derived from the global diversity dataset; after filtering, 849 (4.8% of 17,607) genes were identified with a dN/dS greater than 2."

Could Doyle et al. please add three Supplementary Tables that list these sets of 268, 108, and 849 genes, respectively? All three sets are potentially of considerable interest to other researchers analysing parasitic nematode genomes; but, as things stand (with no supplementary gene lists available), it will be needlessly laborious and time-consuming for those other researchers to have to independently compute what those gene sets actually contain.

Response to review comment:

We agree with the reviewer that these data should have been included. As discussed in response to another reviewer's questions, we have removed the dN/dS analysis from the manuscript, and therefore, have not provided the gene lists associated with those analyses.

To address this comment, we have generated Table S16 which now includes the 269 genes within the top 1% F_{ST} regions.

Note: I am aware that one URL given by Doyle et al.

(<https://biit.cs.ut.ee/gplink/l/HKI5aHzqTa> [biit.cs.ut.ee]) leads to a g:Profiler page that includes a list of the 268 genes. However, this will not necessarily be obvious to the readers, and it does not provide the permanent availability that adding a supplementary table listing the 268 genes will have. Moreover, there is no corresponding Web link for the sets of 108 or 849 genes at all.

Response to review comment:

As above, we have now included Table S16 listing the genes identified and subsequently used in this analysis.

Page 24:

"...largely describing proteolytic and cysteine-type peptidase activity. Analysis of dN/dS from alignments of 9,970 one-to-one orthologs..." This passage might be more readable if the authors began a new paragraph at the word "Analysis..."

Response to review comment:

Agreed and corrected as suggested. As suggested above, we have now removed the dN/dS analyses and text associated with them.

"We did, however, identify a number of genes likely associated with host/parasite interactions...": this might be clearer if it were written as "Among the set of 108 genes, however, we did identify a number of genes that are likely associated with host/parasite interactions..."

Response to review comment:

Agreed and corrected as suggested. As suggested above, we have now removed the dN/dS analyses and text associated with them.

"...were among the most abundant pfam domains." This comes after a list of orthologs of *C. elegans* genes, i.e., after a list of specific *genes* rather than a list of protein domains. Perhaps Doyle et al. could better rewrite this as "...were among the most abundant categories of gene types." or something of that sort. Moreover, even if they wish to stick with the current wording, "pfam domains" should be written "Pfam domains" (i.e. with a capital 'P' in 'Pfam').

Response to review comment:

We acknowledge the potential confusion here in the original version of the text.

As suggested above, we have now removed the dN/dS analyses and text associated with them.

Page 36:

"As such, only short-read RNAseq data were used..." might be better written as "Accordingly, only short-read RNA-seq data were used..." (or "Therefore...").

Response to review comment:

Agreed and corrected as suggested.

Page 38:

"For each transcript, differential expression between life-stages was tested using a Wald test..." This raises an issue with analyzing gene expression with kallisto. The default mode for kallisto analysis is to distinguish expression between *transcripts*, not *genes*; and these are not the same (a gene can have two or more transcripts through alternative splicing). Since sleuth was designed to work with the outputs of kallisto, this default mode of kallisto operation affects sleuth as well. However, the supplementary data tables make it clear that Doyle et al. mapped the kallisto and sleuth results to *H. contortus* genes at some point (in particular, Supplementary Table S8 shows kallisto/sleuth results for transcripts, but then somehow these results were mapped to genes). How was this done? The mapping may have been simple, but I would still like the authors to explain and clarify this point.

Response to review comment:

The reviewer is correct in that the default mode for kallisto analyses is to analyse transcript expression. This is what has been done, and the differential expression of transcripts between pairwise lifestages is presented in Table S9. The transcript identifier is shown by the "-00001, -00002 etc".

The gene set analysis was, however, performed on gene IDs rather than transcript IDs. This is simply because the gProfiler (1) works on Gene IDs, rather than transcript IDs, and (2) after removing the transcript ID extensions, gProfiler removes redundant IDs, ie. if two gene IDs are provided because there are two transcripts, one ID will be removed. The outcome of this will be an underrepresented set of transcripts on which the enrichment analysis was performed. While not ideal, it is at least conservative.

To address this comment, we have made an additional note in Table S9 explaining this difference:

"4. The gene set analysis was performed on genes IDs, rather than transcript IDs, which were incompatible with gProfiler. This was achieved by simply removing the transcript identifier. This accounts for the difference between the gene and transcript counts, and why the gene set enrichment analysis was only performed on the genes. The outcome of this difference will potentially result in a conservative underestimate of transcript enrichment if multiple transcripts from a single gene are differentially expressed between life stages."

Page 41:

"[dN/dS] ratios greater than one indicate positive selection for new protein-coding changes, ratios close to zero indicate neutral evolution, and ratios less than one indicate purifying selection." Mostly this makes sense, but the part about 'ratios close to zero indicate neutral evolution' strikes me as being badly wrong; I think it should read 'ratios close to *one* indicate neutral evolution'. If I am wrong, can the authors please explain why? If I am right, can the authors please correct this error?

Response to review comment:

Thank you for picking this up. The text has now been corrected from “zero” to “one”.

Legend of Supplementary Figure S1:

"we present the changes in the assembly overtime" should read "we present the changes in the assembly over time".

Response to review comment:

Agreed and corrected as suggested.

Figure S4:

In some sense, the division of syntenic gene pairs into the categories of "conserved order" and "conserved order but reorientated" must be arbitrary; as the authors indicate in the main text, although the six chromosomes of *H. contortus* are clearly homologous, they preserve no long-range synteny whatsoever (as shown in Figure 1B). The initial decision about which way to orient a given chromosome (e.g., chromosome I) of *H. contortus* with respect to *C. elegans* therefore must have been done at random. That being said, the nearly perfect split between fully conserved versus conserved-but-reorientated syntenic pairs (Supplementary Figure S3C) is what one would expect given extensive shuffling of microsyntenic genomic regions.

Response to review comment:

The reviewer is correct in that the categories are arbitrary - they were defined simply to describe the different scenarios in which gene pairs could be arranged (or rearranged) between the two genomes. This analysis was performed to begin to provide some resolution towards understanding chromosome structure conservation, given we have chromosome-scale scaffolds to work with - unfortunately, there has been sufficient evolutionary time between *C. elegans* and *H. contortus* to completely reshuffle the order of the genomes. However, this analytical approach provides interesting results for much more closely related species, for example, between chromosomally resolved species within the *Caenorhabditis* genus, and will provide more insight for parasitic species including *Haemonchus* once additional Strongylid genomes are resolved.

Legend of Supplementary Figure S7:

"transcripts with significant but shared differential expression": Although I share the authors' admiration for the clust program and use it for my own work, it does not provide statistical estimates (such as p-values) for the nonrandomness of the expression clusters which it generates. I would thus reword this to something like "transcripts with shared differential expression".

Response to review comment:

Agreed and corrected as suggested.

"putative co-regulated transcripts" should read "putatively co-regulated transcripts".

Response to review comment:

Agreed and corrected as suggested.

Figure S9: This is excellent! It is the first time I have seen multipopulation PCA diversity maps presented in a way that makes it easy for the reader to identify each population in the map (by giving many different maps, one for each distinct population, in which that one population is marked while all other data points are kept constant).

Response to review comment:

We thank the reviewer for appreciating this. While originally made to simply explore the datasets, they are quite useful to help the reader interpret the primary dataset when many groups of many data points are presented in a single plot.

Supplementary Table S2:

'Data columns B and C are both labeled "H. contortus V4", which is redundant and confusing. I believe the authors may have meant "H. contortus V4 Chromosomes" (column B) and "H. contortus V4 Haplotypes" (column C), which would make much more sense, and which would be consistent with data column names used in Supplementary Table S3. Please review these column names and (if possible) correct them so that they are nonredundant and nonconfusing.

Response to review comment:

We completely agree - there seems to be a reformatting issue converting between google sheets (where it is correct as the reviewer has suggested) and excel for submission (where it is not correct).

We will ensure the correctly formatted supplement is resubmitted.

Supplementary Table S5:

The "Precision" values for gene models from the authors' previously published V1 genome assembly, in this table, look astoundingly low. They are in the low single digits (or even low *single decimals*) for every precision value: base, 2.7; exon, 2.6; intron, 2.7; transcript level, 0.7 [!]; and locus level, 0.8 [!]. In contrast, the precision values for all of the V4 gene models are much more sane (ranging from 29.8 to 98, with most values above 60). Are the V1 precision values really that bad, or is this some sort of systematic typographical error?

Response to review comment:

Thank you to the reviewer for picking up on this. It was clearly incorrect, and we missed this in revision before submission.

In response to this, we have rerun our gffcompare analysis of the V1 annotation vs V1 curated geneset, and updated the precision values which are now much more sensible.

Supplementary Table S6:

Data column C is labeled simply "H. contortus", which is confusing. I believe the authors may have meant "H. contortus V1" (i.e., their 2013 genome assembly which was the precursor for their V4 assembly now). Please review this column name and (if possible) correct it so that it is nonconfusing.

Response to review comment:

We believe this is the same reformatting error as above. We will correct this for resubmission.

Supplementary Table S8:

"Treshold for DE was $q_value < 0.01$ and $b < -1$ or $b > 2$ " should read "Threshold for DE was $q_value < 0.01$ and $b < -1$ or $b > 2$ " (i.e., correct the spelling of 'Threshold').

Response to review comment:

Agreed and corrected as suggested.

Reviewer #2 (Remarks to the Author):

I find this a really good manuscript overall. The analysis is largely based on a good quality de novo genome assembly/annotation and the data and analyses are extremely high quality - in many ways I think this will be used as a model for many future papers on a diverse range of species.

I was initially slightly skeptical (before having read the full supplement) that the results might have been affected by potential issues derived from the fact that the genomes were not all generated in a single go. However, it's clear from the supplement that the authors have taken considerable steps to account for this, and thus I have no technical concerns at all.

I have only few minor comments:

1) CEGMA

CEGMA is discontinued, there is no need to use this, unless there is a specific feasible justification not reported here. The CEGMA's authors, strongly recommend do not use this tool anymore: "We suggest you consider using alternative tools". The last update was in July 30th 2014. <http://korflab.ucdavis.edu/datasets/cegma/>. [korflab.ucdavis.edu]

As I have mentioned before, this paper has an enormous potential to be used as a model for future studies. I truly believe CEGMA analyses should be removed in this paper.

I suggest to keep only the information from BUSCO.

Response to review comment:

We acknowledge the reviewer's comment regarding use of CEGMA, and that the reviewer points out that CEGMA's authors discourage its use. We are aware of this statement on the CEGMA website. However, we and others have anecdotally noticed that BUSCO does not always perform reliably for nematode genomes, evidence of which is shown in Additional File 1. CEGMA, on the other hand, does tend to perform reliably.

There is an assumption that because the tool is no longer supported, it does not work, or that because a new tool comes along, it is therefore better. However, prior discussions with colleagues and even with CEGMA's author Keith Bradnam has convinced us that presenting the output of both tools provides useful information and so we have decided to keep it.

2) Synteny

Pag 7: “Complete within-chromosome synteny would be expected to show a positive linear relationship between the genomic positions of orthologs between *C. elegans* and *H. contortus* chromosomes; “

“We show a remarkable pattern of almost complete conservation of chromosome content (synteny) with *Caenorhabditis elegans*, but almost no conservation of gene order.”

Here, I really missed a reference to the statements. A reference to the evolutionary distance between both species and why higher conservation in the gene order was expected. We know that theoretically two orthologs genes should share homologous neighboring genes, however, there is a chance of non-homology matches occurring in collinearity.

Probably the authors are trying to tell a different history from the previous paper:<https://www.ncbi.nlm.nih.gov/pmc/articles/PMC4054779/pdf/gb-2013-14-8-r88.pdf> [[ncbi.nlm.nih.gov](https://www.ncbi.nlm.nih.gov/)], but they cannot ignore the referential for the evolution. Also, looking for this previous work I would not expect a higher gene order conservation between *C. elegans* and *Haemonchus contortus*.

Also, how can Trans and Cis splicing can affect the gene order, as it has been extensively described in *C. elegans*.

Response to review comment:

We believe that we have not communicated this clearly, and/or the reviewer has missed our intention with some of the statements highlighted above. We have not intentionally tried to “tell a different history” from the Laing et al 2013 publication, but will try to explain how and why we have made the above statements.

“Complete within-chromosome synteny would be expected to show a positive linear relationship between the genomic positions of orthologs between *C. elegans* and *H. contortus* chromosomes; “

...was used to suggest that IF the chromosomes of these species were completely syntenic, then there would be a positive, linear relationship between the positions of shared orthologs between the two genomes. This is a hypothetical scenario but was provided to suggest to the reader to help interpret the figure. We do follow this statement, linked by a semi-colon, that states:

“this is not observed, with an almost complete reshuffling of orthologs between the two species.”

This is again reiterated in the text:

“We show a remarkable pattern of almost complete conservation of chromosome content (synteny) with *Caenorhabditis elegans*, but almost no conservation of gene order.”

... was the result, where almost no synteny was observed, i.e., the complete opposite of the linear relationship was observed, and hence, we conclude there was very little synteny. We then go further to quantify this and explore microsynteny in some more detail.

The whole point of this is to emphasise that we did not expect a lot of synteny - this had already been described from analyses of genome fragments previously - but that (i) a circos plot provides a visual overestimation of synteny because it lacks resolution, and (ii) because we now have chromosomes, we can describe genome-wide synteny in more detail. As mentioned above, we have gone to the extent of quantifying this synteny (or lack of) using novel approaches. As more chromosome-scale genome assemblies become available, these types of analyses will become more relevant and commonplace.

Regarding how trans- and cis- splicing may affect gene order; we are unaware of how cis splicing may result in gene order conservation, however, trans-splicing of genes will likely result in gene order conservation due to the maintenance of operons. We have speculated that this is the case for microsyntenic conservation, where we state conservation is:

“likely representing selective constraint to maintain evolutionarily conserved operons,”

We do discuss some of the similarities and differences (with relevant references) in transplicing between *H. contortus* and *C. elegans* later on in the manuscript.

To address this comment, we have amended the first statement as follows:

“Hypothetically, if *C. elegans* and *H. contortus* chromosomes were completely colinear, the genomic positions of orthologs would show a positive linear relationship between the two species; however, this is not the case, whereby an almost complete reshuffling of orthologs was observed.”

3) Figure 2. Haplotype and repeat distribution within and between chromosomes
That’s a really bad legend. Most part of the text should be explained in the main text.
Also, the Chrx (A,B) is quite confuse. The authors need to add one more letter (A,B,C,D) to explain this part.

Response to review comment:

In response to the comment, we have endeavoured to improve the description of the figure legend to make it clearer to the reader.

We will discuss the suitability of the length of the legend with the Editor.

Same for the Figure5. The legend contains a lot of unnecessary information.

Response to review comment:

Similar to the comment above, we have reviewed the wording of the legend and will discuss the length with the Editor.

4) Pag12:

Generation of a high-quality transcriptome annotation incorporating short and long reads

This is an elegant strategy. Could you please check the correct Apollo reference (Ref: 37)? I believe that you have used the WebApollo version (page: 37), if this is the case, please correct the reference.

WebApollo:

<https://genomebiology.biomedcentral.com/articles/10.1186/gb-2013-14-8-r93>
[genomebiology.biomedcentral.com]

Response to review comment:

We thank the reviewer for appreciating our approach to generate the annotation.

Regarding the citation of Apollo, we have an in-house instance of Apollo to visualise and curate the *Haemonchus* annotation, which was setup and is maintained in collaboration with the lead developer Nathan Dunn and using documentation available here: <https://genomearchitect.readthedocs.io/en/latest/>. As WebApollo and Apollo have become integrated into a single product, the recommended paper to cite by the developer and as stated in the documentation is the one we have cited.

5) Figure S6. Annotation pipeline schematic used to incorporate RNAseq and IsoSeq data into a single annotation. The picture is in a low resolution. My reading was somewhat impaired.

Response to review comment:

We acknowledge that the resolution decreased when converting to a PDF, and will seek to improve this for resubmission.

6) RATT

Figure S6. Initial attempts to transfer genome annotations from the V1 assembly to the intermediate V3 assembly using tools such as Ratt [87] were unsuccessful, and with the generation of new Iso-Seq data, a new strategy was designed and implemented.

It's unclear to me (on page 36) what exactly was done with the RATT. If it was just an unsuccessful transfer attempt, I believe this should not be mentioned.

The authors may decide to remove this information.

Response to review comment:

The reviewer is correct in that we attempted to use RATT to transfer gene models from V1 to our improved assembly, but that this was unsuccessful. We included this statement to highlight that we tried and failed; however, gene model transfers between assembly versions are not trivial, and as we go on to describe, we generated new data that required a new strategy to annotate the genome.

In response to this comment, we have removed reference to RATT.

7) GAG

Was this tool used only to generate the annotation metrics? If so, it is ok!

If this tool has been used for some type of correction of the annotation, I am very distressed about the possible results, it may have implications for their quality of the annotation. This is a really bad and dangerous tool that unfortunately has been used a lot, but it shouldn't!

If the authors have another tool to reach the metrics, I strongly recommend not to mention the use of GAG.

Response to review comment:

We thank the reviewer for the comment. GAG was only used to calculate the annotation metrics as described in the methods, and was not used for any other part of the annotation process.

8) Genome Annotation – Manual Curation

I remain in doubt: Was a manual curation of the genome generated?

If it was done, it is necessary to add a line paragraph of how it was generated.

No more comments.

Response to review comment:

There was relatively minimal manual curation performed for the assembly version presented in the manuscript. A small subset of genes was curated to provide sensitivity and specificity estimates on the assembly, as described in the results and supplementary tables. Further curation was performed in a subset of genomic regions that were the focus of other work that is not relevant to this manuscript. As described in a previous comment above, we have an in-house Apollo instance to manually curate the genome and a mechanism to provide updates as necessary via WormBase ParaSite. The data used for this is described in the methods section, and the status of the genome annotation is declared in the “Availability of data and materials”, which states:

“A static version of the genome annotation used in this paper is available at ftp://ftp.sanger.ac.uk/pub/pathogens/sd21/HCON_V4_GENOME/ (signoff date: 25th Jan 2019), however, the most up-to-date version of the annotation can be accessed at WormBase ParaSite (https://parasite.wormbase.org/Haemonchus_contortus_prjeb506/Info/Index/; (Howe et al., 2017)).“

This, therefore, allows the reader to access the annotation directly relevant to the manuscript from the FTP, however, the most up-to-date annotation should be accessed from WormBase Parasite.

Reviewer #3 (Remarks to the Author):

The manuscript by Doyle et al. presents a chromosome-scale assembly of the parasitic nematode *Haemonchus contortus*. This represents a major advancement over previously published genomes which suffered from the high levels of genetic diversity in the sequenced population that caused overestimated genome sizes, substantial misassembly, and false signals of duplications. The haplotype resolved assembly together with new Iso-seq and RNA-seq data allowed the authors to generate a vastly improved set of gene annotations and to use this new annotation to reinvestigate the developmental transcriptome and trans- and cis-splicing in this parasitic nematode. Furthermore, they provide an update on the population genomics of this species using newly sequenced isolates. Altogether, this manuscript represents a massive and highly important amount of work that definitely deserves publication.

Response to review comment:

We thank the reviewer for their positive appraisal of our work.

I only have some minor comments.

abstract:

"The *H. contortus* MHco3(ISE).N1 genome assembly presented here represents the most contiguous and resolved nematode assembly outside of the *Caenorhabditis* genus to date," -> What about *Strongyloides ratti* and *Pristionchus pacificus*? I would prefer a more modest term such as "represents one of the most contiguous". Also, even if the statement is true, for how long will it hold?

Response to review comment:

We agree with the reviewer that a more modest statement perhaps has better longevity, and so have changed the text as suggested. We have also modified other parts of the manuscript along these lines so that it doesn't become wrong when more contiguous assemblies hopefully appear soon.

p. 3

"smallest of genomes or for select organisms, where genome assembly projects"
-> "selected organisms" ?

Response to review comment:

Agreed and corrected as suggested.

p. 3

"the excessive diversity presents a significant challenge to the assembly process, and consequently, typically results in fragmented assemblies that contain misassembled and haplotypic sequences."

I think Barriere et al. *Genome Res.* 2009 Mar;19(3):470-80. is an important work that deserves to be cited here.

Response to review comment:

We thank the reviewer for this suggested reference. We agree that this is an excellent example of the challenge described, and so have cited this paper.

p. 6

I do not like the term "recombined gene pairs" because the link to the biological process of recombination is very loose. I would prefer something like "rearranged", "reshuffled".

Response to review comment:

We acknowledge the reviewers' concern and agree that it could be interpreted in more ways than what is intended.

To address this comment, we have changed "recombined" with "rearranged".

Legend Figure 1

a) "Complete within-chromosome

synteny would be expected to show a positive linear relationship between the genomic positions of orthologs between *C. elegans* and *H. contortus* chromosomes;"

-> "Complete within-chromosome collinearity" ?

Response to review comment:

Agreed and corrected as suggested.

b) Check that species names are in italic.

Response to review comment:

Agreed and corrected as suggested.

c) The lower density of genes in the center of the X chromosome is so striking that it should be commented on.

Response to review comment:

We agree with the reviewer that this should be commented on.

To address this comment:

- In the results, we have provided a statistical test of the gene density and gene coverage between autosome and X chromosome
- In the discussion, we have referred to the lower gene density and gene coverage as a feature of difference between the autosomes and X chromosome.

d) Given that there are around 1000 dots on each of the plots, I am not sure if one can visually detect a positive linear relationship even if it existed. I wonder, how these plots would look like with the smoothScatter function in R that displays the density of data points.

Response to review comment:

We agree that this figure is a little “busy” and is difficult to determine correlated relationships. However, this is the point, as there are very few. We go into some more detail defining this further along in the manuscript. Note these plots are in complete contrast to similar plots generated between more closely related species, for example within the *Caenorhabditis* genus (unpublished data), or between tapeworm genomes (Olson et al 2020 bioRxiv <https://doi.org/10.1101/2020.04.08.031872>) in which the relationships between chromosome, or regions of chromosomes, are much more evident.

As each point does refer to a defined position in each genome, and density or binning based reduction would remove this positional information, we would prefer to keep these plots.

p. 8

"high average orthologs per partial and complete CEGMA genes"

What does "average orthologs" mean?

Response to review comment:

Average orthologs refers to a metric produced by CEGMA to describe the number of hits per CEGMA gene found in the assembly. The expectation is that there is a single ortholog in the genome per CEGMA gene; values above 1 indicate duplications that may be erroneous.

In response to this comment, we have modified the wording to the following:

“...and the increased proportion of orthologs detected, on average, per partial and complete CEGMA gene...”

p. 8

" When compared against publically available genomes in WormBase ParaSite, the V4 genome assembly represents the most contiguous and resolved nematode assembly outside of the *Caenorhabditis* genus, and second-most complete Clade V parasitic nematode assembly based on the presence of universally conserved orthologs (n = 20; *Ancylostoma ceylanicum* is the most complete based on BUSCO analysis; Additional file 1: Figure S4 A,B).

"  First, contiguity is usually measured in N50 (for example the *P. pacificus* assembly has a contig N50 of 4.6 p.7 (Rödelsperger et al. 2017), vs *Haemonchus* 3.8 Mbp p.7), Additional file 1: Figure S4 A,B only shows BUSCO data. Second, it is not obvious that *Haemonchus* is the most complete genome among the tested ones in Figure S4 A,B.

Response to review comment:

We agree with the reviewer’s comment regarding contiguity and clarity around the BUSCO data. In line with our response to one of the reviewers initial comments, we have chosen to

remove this statement as “hopefully” our assembly will be superceded but more contiguous assemblies in the near future. We have also chosen to remove Figure S4B - we initially included this to show the distribution of BUSCO and CEGMA scores, and highlight where *Haemonchus* was placed relative to this distribution, however, given the reviewers comment, we don't think it really adds to the narrative of the manuscript.

Fig 2C

- a) this panel is far too small, I would recommend to split this figure into two.
- b) how does repeat count (number of repeats) compare to repetitive fraction (Amount of sequence that is covered by repeats). In terms of gene density, people often use different terms which can easily cause misunderstanding.

Response to review comment:

We agree with the reviewer that Figure 2 C is quite small and densely populated. However, it illustrates genome-wide trends rather than specific details, and splitting it would only increase page space without revealing new information. We are happy to be convinced otherwise, but think it should remain as it is.

We also agree with the reviewer that there is a difference in the interpretation of counts vs coverage per window and that it has the potential to cause some confusion. We have used both - in Figure 2C we show repeat count per Mbp, and in Figure S5, we have used coverage per 200 kbp. Once again, these plots show broad, very zoomed-out genome-wide trends, however, comparison of the peaks and troughs show that they are quite concordant.

p. 13

It would be nice to compare how the improved annotation is reflected at the BUSCO level.

Response to review comment:

We agree with the reviewer that it would be good to use BUSCO to compare the annotation improvement.

In response to this comment, we have extended the Table S6 that showed sensitivity and specificity of the assembly improvement to now also include the BUSCO scores generated for the translated protein sequences for each annotation stage presented.

Fig 3b Why is the feature count so important? Would not just showing the length distributions of genomic features in different haemonchus annotations be most informative?

Response to review comment:

The feature count is not necessarily important, but does provide a somewhat informative variable to use in a scatterplot, which is the most concise way to show these data for 5 feature classes and 4 annotations.

We originally intended to show length distributions in the supplement, however, they take up more space with less information, and because they tend to have long tails in the distribution, the main bulk of the data was not as easily distinguished between species if all data are included.

Unless there is a strong objection, we prefer to keep this scatter as a simple overview.

p 18

"and sequence conservation in the genome sequence" avoid to use "sequence" twice?

Response to review comment:

Agreed and corrected as suggested.

Fig 5B does NA refer to cases truly without splice leaders or ambiguous genes that are partially transplined to SL1 and SL2? Are there examples of that?

Response to review comment:

We use NA to describe genes without SL1 or SL2 sequences, at least as defined by our parameters used to detect them. We have now added this to the figure legend. It is almost certain that deeper transcriptome sequencing will reveal a greater proportion of genes that do have SL sequences, however, this sequencing is outside the scope of the current work.

Fig 5D females and males should be labeled.

Response to review comment:

Agreed and corrected as suggested.

Fig 5E What is the source of the gene models below the RNA-seq tracks

Response to review comment:

We are not entirely sure that we are addressing the reviewer's question, however, the "source of the gene models below the RNA-seq tracks" is from the *H. contortus* annotation described throughout, and are visualised using Apollo.

p 19.

"Although these variations have been described in *C. elegans* [47], they seem to be more of the norm rather than the exception in *H. contortus*."

I think Sinha et al. RNA. 2014 Sep;20(9):1386-97. found similar patterns. This reference could be taken to support that this observation might not simply an exception in a weird parasite.

Response to review comment:

We agree with the reviewer that the Sinha 2014 paper suggested also shows variant patterns of SL1/SL2 trans splicing when compared with *C. elegans* as we have done with *H. contortus* here. Here, we have described specific observations in our analysis of *H. contortus* and said that these differences (compared with *C. elegans*) are more common than what are considered exceptions to the general rule in *C. elegans*. So, while we agree that *P. pacificus* also has differences with *C. elegans*, and very likely supports that "this observation might not simply an exception in a weird parasite", the phrase here is not describing the norm in nematode genomes broadly, but across operons within the *Haemonchus* genome, so including this reference does not quite fit with the intended message communicated.

p 24

"We extended this analysis to *H. contortus* variation derived from the global diversity dataset; after filtering, 849 (4.8% of 17,607) genes were identified with a dN/dS greater than 2." $dN/dS > 2$ sounds very suspicious. This could be caused by misestimations in cases of very low overall divergence or very short proteins, probably also other causes like misalignments. Anyway, this screen for adaptation is highly speculative and in my opinion distracts from the main selling point of the paper (see general comments).

Response to review comment:

We included these analyses to begin to explore putative evidence of selection on shaping the genome wide distribution of genetic variation as we demonstrate in Figure 6 and associated description. We don't necessarily agree that this distracts from the main selling points of the paper, but it is somewhat of a tangent; we do, however, we agree with the reviewer that this section could be more robust and perhaps needs to be expanded upon to fully explore the data.

In response to this comment, and, given we also agree with the reviewer's next comment that the manuscript is "pretty long anyway", we have decided to remove this analysis and associated description of the data.

General comments:

Some sections such as the Gene Ontology analysis of expression clusters are very descriptive. Given that the manuscript is pretty long anyway, I would appreciate if the authors could stronger focus on the key findings of every analysis. For example, the population genomic analysis should stronger focus on patterns that could not have been characterized with previous versions of the *H. contortus* assembly and the new population data from Pakistan. I do not see why the UK vs French comparison is of importance here (p.22).

Response to review comment:

This description of the data is simply to provide a stage for introducing the new data on the new genome (particularly as a resource that is publicly available), and to provide context for the genetic placement of the reference strain used in the genome assembly. This is not dissimilar to many published papers that place new sequences samples in the context of available data. Again, in agreement with the reviewer that the manuscript is "pretty long anyway", we don't believe this is the place for a detailed population genetic analysis.

We have however taken on board the comment that we could focus on the key findings of every analysis, and have amended the text in relevant places where we believe this is needed.

A lot of space in Figures is taken up by examples. I am not sure to what extent these examples are needed. These arrangements should be reconsidered.

Response to review comment:

We acknowledge the reviewer's concern that "a lot of space in Figures is taken up by examples", however, we argue that these highlight important concepts that help the reader to understand following aspects of the analysis.

We are happy to be convinced otherwise, and will discuss with the Editorial team on this matter.

Figure captions can be streamlined by adding proper legends to individual panels. e.g Fig 4c, Fig 2C,D

Response to review comment:

Similar to the above comment, we will discuss with the Editorial team regarding whether the figure legends require streamlining.

The authors compare version V1 and V4 of the *Haemonchus* annotations, a comment on V2 and V3 would be much appreciated.

Response to review comment:

This is a very sensible comment to make, however, we did not annotate V2 and V3 assemblies.

To address this comment, we have included the following text in the methods:

“Annotation statistics were determined using *Genome Annotation Generator (GAG)* (Geib et al., 2018). Annotation comparison was only made between the previously published V1 draft assembly (Laing et al. 2013), the final V4 chromosomal assembly presented here, the McMaster (Schwarz et al. 2013) and NZ isolate (Palevich et al. 2019) draft assemblies, and the *C. elegans* reference; no comparable annotation was generated for the intermediate assemblies generated before V4 was finalised, including assembly versions V2 and V3.”

The authors should avoid citing manuscripts that have not gone through peer review, e.g. biorxiv. Biorxiv is a nice platform to put ideas and to provide early access to new results, but these articles should not be cited if their results have not at least been approved by reviewers.

Response to review comment:

We cite the following manuscripts that have been submitted to recognised preprint servers (arxiv / bioRxiv), which is accepted practice by Communications Biology as described in the instructions to authors.

- Li, H. (2013). Aligning sequence reads, clone sequences and assembly contigs with BWA-MEM.
- Nattestad, M., Chin, C.-S., and Schatz, M.C. (2016). Ribbon: Visualizing complex genome alignments and structural variation.
- Stewart, R.D., Auffret, M.D., Warr, A., Walker, A.W., Roehe, R., and Watson, M. (2018). The genomic and proteomic landscape of the rumen microbiome revealed by

comprehensive genome-resolved metagenomics

- Wintersinger, J., Mariene, G.M., and Wasmuth, J. (2018). One species, two genomes: A critical assessment of inter isolate variation and identification of assembly incongruence in *Haemonchus contortus*.
- *** New: Olson, P.D., Tracey, A., Baillie, A., James, K., Doyle, S.R., Buddenborg, S.K., Rodgers, F.H., Holroyd, N., and Berriman, M. (2020). Complete representation of a tapeworm genome reveals chromosomes capped by centromeres, necessitating a dual role in segregation and protection.

In response to this comment, we have:

- updated the Stewart et al 2018 citation to the current Nature Biotechnology 2019 publication - the title changed so we missed the published version unintentionally.
- Kept the Li 2013 arxiv citation - it is a relevant manuscript to cite for BWA mem that was never published.
- Kept the Nattestad et al 2016 citation - the citation acknowledges the use of software for visualisation, Genome Ribbon.
- Kept the Wintersinger et al 2018 citation - it provides a critical analysis of the two draft *Haemonchus* genomes published in 2013, highlighting discrepancies between the two. Our manuscript describes the improvement of one of those genomes. The Wintersinger bioRxiv manuscript not only provides insight into the differences between genome assemblies of the same organism but also adds weight to the rationale to improve draft genomes to better understand and characterise the genome biology. Communication with the senior author of the Wintersinger manuscript revealed that given our genome was significantly improved, their manuscript is now somewhat outdated and therefore publication was not going to be pursued. However, we believe this is still a relevant citation to use our manuscript.
- Added the bioRxiv preprint to the chromosome-scale assembly of *Hymenolepis microstoma*. This was referred to in the introduction already as being available at WormBase Parasite but not in the literature, however, it has recently been submitted to bioRxiv and for peer review.

Reviewers' Comments:

Reviewer #1:

Remarks to the Author:

I have reviewed the revised manuscript by Doyle et al., and am fully satisfied with their revisions and their responses to my previous comments.

I have only two further minor revisions of their revised manuscript to suggest:

Lines 715-716: "...has been described previously in both parasitic and the free-living Clade V nematodes..." might better read as "...has been described previously in both parasitic and free-living Clade V nematodes..." (i.e., delete 'the' before 'free-living').

Figure 2c: Although I think I can *guess* which of the six chromosomes are chromosomes I through V and X, it would be better if the article's readers were not required to guess. Could the authors please add chromosome numbers (I, II, III, IV, V, and X) next to the outer edge of each circular chromosome arc, so that their identities are obvious?

Reviewer #3:

Remarks to the Author:

Doyle et al. have done a really good job on addressing my previous concerns. The manuscript is now ready for publication.

Response to reviewer comments

Reviewer #1 Remarks to Authors:

I have reviewed the revised manuscript by Doyle et al., and am fully satisfied with their revisions and their responses to my previous comments.

Response to review comment:

We are happy to see we have addressed the reviewers comments satisfactorily.

I have only two further minor revisions of their revised manuscript to suggest:

Lines 715-716: "...has been described previously in both parasitic and the free-living Clade V nematodes..." might better read as "...has been described previously in both parasitic and free-living Clade V nematodes..." (i.e., delete 'the' before 'free-living').

Response to review comment:

Agreed, and updated as suggested.

Figure 2c: Although I think I can *guess* which of the six chromosomes are chromosomes I through V and X, it would be better if the article's readers were not required to guess. Could the authors please add chromosome numbers (I, II, III, IV, V, and X) next to the outer edge of each circular chromosome arc, so that their identities are obvious?

Response to review comment:

Agreed, and updated as suggested.

Reviewer #3 (Remarks to the Author):

Doyle et al. have done a really good job on addressing my previous concerns. The manuscript is now ready for publication.

Response to review comment:

We thank the reviewer for their appraisal, and appreciate their support of our manuscript for publication.

Editorial Team:

1) Due to our policy on Transparent Peer Review, you must clearly state in your cover letter for manuscripts accepted in principle whether you wish to "OPT IN" or "OPT OUT" to the publication of the reviewer reports.

Response to Editorial comment:

Completed as suggested. We have chosen to OPT IN.

2) Please note that the item you have currently titled Supplementary Data 1 should be titled Supplementary Information. Please then title Supplementary Data 2 as Supplementary Data 1.

Response to Editorial comment:

Completed as suggested.

3) Please note you have cited Supplementary Tables but they are not provided in the Supplementary Information file. Supplementary Tables cannot be provided outside of the main Supplementary Information file. If you wish to provide separate files, they will need to be Supplementary Data 1, Supplementary Data 2 etc instead.

Response to Editorial comment:

Completed as suggested.

4) If you provide Supplementary Data files, please ensure that you provide legends for these in the Author Cover Letter.

Response to Editorial comment:

The supplementary data file headings have now been included in the cover letter.

Editor:

-Please retain Supplementary Tables 1, 2, 9, 11, 12, 13, 15, 16 in the Excel file and refer to these as Supplementary Data 1, Supplementary Data 2, etc in both the Excel file and the main text document. Please move the remaining tables into the Supplementary Information pdf and re-number them sequentially.

Response to Editorial comment:

Completed as suggested.

-Please change "lead" to "led" in the Author Contributions statement.

Response to Editorial comment:

Completed as suggested.

-We ask that code is also deposited in a doi-minting repository such as Zenodo and included in the reference list and referenced in the code availability statement.

Response to Editorial comment:

Completed as suggested.